# Quantum Optimization via Gradient-Based Hamiltonian Descent

**Jiaqi Leng** [1] [2]   **Bin Shi** [3] [4]

## Abstract

With rapid advancements in machine learning, first-order algorithms have emerged as the backbone of modern optimization techniques, owing to their computational efficiency and low memory requirements. Recently, the connection between accelerated gradient methods and damped heavy-ball motion, particularly within the framework of Hamiltonian dynamics, has inspired the development of innovative quantum algorithms for continuous optimization. One such algorithm, Quantum Hamiltonian Descent (QHD), leverages quantum tunneling to escape saddle points and local minima, facilitating the discovery of global solutions in complex optimization landscapes. However, QHD faces several challenges, including slower convergence rates compared to classical gradient methods and limited robustness in highly non-convex problems due to the non-local nature of quantum states. Furthermore, the original QHD formulation primarily relies on function value information, which limits its effectiveness. Inspired by insights from high-resolution differential equations that have elucidated the acceleration mechanisms in classical methods, we propose an enhancement to QHD by incorporating gradient information, leading to what we call gradient-based QHD. Gradient-based QHD achieves faster convergence and significantly increases the likelihood of identifying global solutions. Numerical simulations on challenging problem instances demonstrate that gradient-based QHD outperforms existing quantum and classical methods by at least an order of magnitude.

[1]Simons Institute for the Theory of Computing, University of California, Berkeley, USA [2]Department of Mathematics, University of California, Berkeley, USA [3]Center for Mathematics and Interdisciplinary Sciences, Fudan University, Shanghai, China [4]Shanghai Institute for Mathematics and Interdisciplinary Sciences, Shanghai, China. Correspondence to: Jiaqi Leng <jiaqil@berkeley.edu>.

*Proceedings of the 42nd International Conference on Machine Learning*, Vancouver, Canada. PMLR 267, 2025. Copyright 2025 by the author(s).

## 1. Introduction

In modern machine learning, a central challenge lies in unconstrained optimization, particularly the task of minimizing a continuous objective function without any constraints. Mathematically, this problem is formulated as:

$$\min_{x \in \mathbb{R}^d} f(x).$$

Efficiently solving such optimization problems is fundamental to a wide range of machine learning applications. First-order optimization algorithms have emerged as the cornerstone of this endeavor due to their computational efficiency and low memory requirements. One of the simplest yet most widely used first-order methods is the vanilla gradient descent, which updates iteratively according to:

$$x_{k+1} = x_k - s\nabla f(x_k),$$

where $s > 0$ denotes the step size. This method, though simple, serves as the foundation for many modern optimization techniques. In the early 1980s, a groundbreaking advancement was introduced by Nesterov (1983): the accelerated gradient method, now widely known as Nesterov's accelerated gradient descent method (NAG). This method revolutionized first-order optimization by achieving a faster convergence rate compared to vanilla gradient descent. The iterative update rules for NAG are as follows:

$$x_k = y_{k-1} - s\nabla f(y_{k-1}),$$
$$y_k = x_k + \frac{k-1}{k+2}(x_k - x_{k-1}),$$

where $s > 0$ is the step size. The key innovation of NAG lies in the introduction of momentum, which effectively reduces oscillations in the optimization trajectory and speeds up progress towards the optimal solution.

Recent advancements have shed light on the mechanisms underlying the acceleration of NAG, thereby effectively bridging the gap between its discrete updates and the continuous dynamics of damped heavy-ball motion. One pivotal contribution in this area is the introduction of the low-resolution ordinary differential equation (ODE) by Su et al. (2016), which characterizes the continuous limit of NAG as:

$$\ddot{X} + \frac{3}{t}\dot{X} + \nabla f(X) = 0,$$

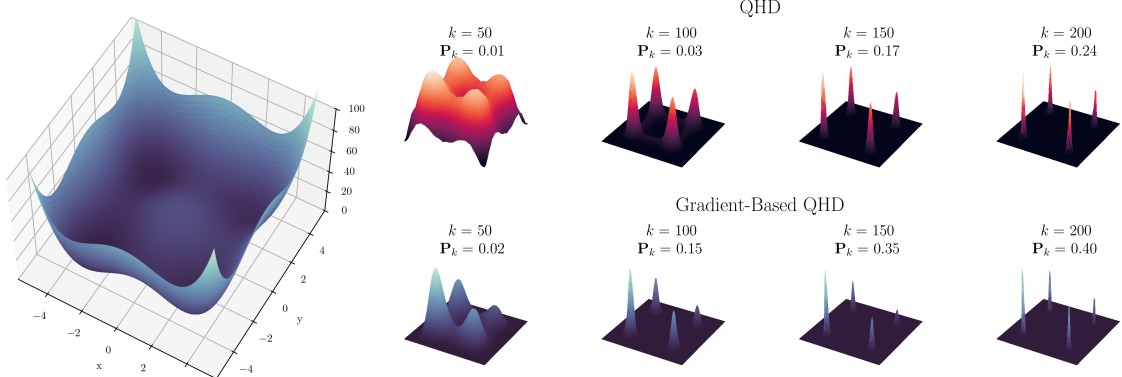

*Figure 1.* Numerical comparison of successful probability across iterations for both QHD and gradient-based QHD applied to the Styblinski-Tang function. $\boldsymbol{P}_k$ denotes the success probability at iteration $k$.

where the first derivative $\dot{X}$ represents velocity in classical mechanics. By transforming this equation into its canonical form, we obtain:

$$\begin{cases} \dot{X} = V, \\ \dot{V} = -\dfrac{3}{t}\dot{V} - \nabla f(X). \end{cases}$$

This canonical form establishes the foundation for a variational perspective on the acceleration phenomenon, which is articulated through the Bregman Lagrangian,

$$\mathcal{L}(X, V, t) = \frac{1}{2}t^3\|V\|^2 - t^3 f(X), \tag{1}$$

as introduced by Wibisono et al. (2016). Furthermore, employing the Legendre transformation, we can convert this Lagrangian into its Hamiltonian form:

$$H(X, P, t) = \frac{1}{2t^3}\|P\|^2 + t^3 f(X), \tag{2}$$

which paves the way for extending the analysis from classical dynamics to quantum dynamics.

By transforming the classical momentum variable $P$ to the quantum momentum operator $-i\nabla$ within the Hamiltonian (2), Leng et al. (2023a) have pioneered a groundbreaking algorithm, known as Quantum Hamiltonian Descent (QHD), which defines the quantum dynamics through the following Schrödinger equation as

$$i\partial_t \Psi(t, x) = \hat{H}(t)\Psi(t, x), \tag{3}$$

where the time-dependent Hamiltonian is articulated as[1]:

$$\hat{H}(t) = \frac{1}{2}\left\|t^{-3/2}(-i\nabla)\right\|^2 + t^3 f(x). \tag{4}$$

---
[1]The original formulation of QHD allows a more general Hamiltonian: $\hat{H}(t) = e^{\alpha_t - \gamma_t}(-\Delta/2) + e^{\alpha_t + \beta_t + \chi_t}f(x)$, where the Laplacian $\Delta = \nabla \cdot \nabla$, as given in Eq. (A.24) of (Leng et al., 2023a). For simplicity, we specialize to the parameter choices corresponding to the classical NAG, namely $\alpha_t = -\log(t)$ and $\beta_t = \gamma_t = 2\log(t)$.

Let $\Psi(t, x)\colon [0, \infty) \times \mathbb{R}^d \to \mathbb{C}$ denote a quantum wave function, whose squared modulus $|\Psi(t, x)|^2$ represents the probability distribution of a hypothetical quantum particle in $\mathbb{R}^d$ at any time $t \geq 0$. For sufficiently large evolution time $t$, the probability distribution is expected to concentrate near the low-energy configurations of the potential $f$, particularly around its global minimum. Measuring the quantum state in the computational basis at such times yields a random vector $X \sim |\Psi(t, x)|^2$, which is likely to lie close to the global minimizer of $f$, thereby approximately solving the associated optimization problem.

As a quantum algorithm, QHD is implemented by simulating the time-dependent Hamiltonian (4), which relies only on oracle access to the function values of $f$. Thus, QHD can be viewed as a quantum *zeroth-order* method. A natural extension of QHD is to develop its higher-order variants that leverage additional information such as the gradient of $f$, and to analyze whether such extensions can enhance QHD's efficiency on various continuous optimization problems.

Inspired by the high-resolution ODE framework introduced by Shi et al. (2022), where the Lyapunov function is conceptualized as a form of energy or Hamiltonian involving the interplay of kinetic energy and gradient, we propose a novel time-dependent Hamiltonian as

$$\begin{aligned} \hat{H}(t) = & \frac{1}{2}\left\|t^{-3/2}(-i\nabla) + \alpha t^{3/2}\nabla f\right\|^2 \\ & + \frac{\beta}{2}\|t^{3/2}\nabla f\|^2 + (t^3 + \gamma t^2)f(x). \end{aligned} \tag{5}$$

In this paper, we mainly investigate the Schrödinger equation (3) with the gradient-based Hamtiltonian (5), termed as **gradient-based QHD**.

## 1.1. Warm-up: gradient-based QHD v.s. QHD

We provide a numerical example to illustrate the differences between gradient-based QHD and standard QHD, both qualitatively and quantitatively. Figure 1 visualizes the probability distribution across iterations for both QHD and

gradient-based QHD, applied to the non-convex Styblinski-Tang function, which features three local minima alongside a global minimum.

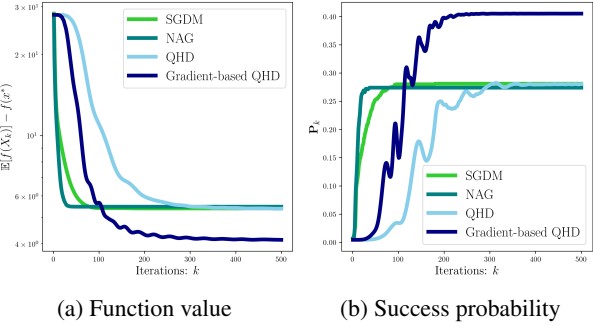

(a) Function value (b) Success probability

*Figure 2.* Numerical performance comparison of various algorithms on the Styblinski-Tang function.

Furthermore, Figure 2 demonstrates the numerical performance involving function values and success probability. While QHD does not depict an obvious advantage against stochastic gradient descent with momentum (SGDM) (Shi, 2024; Shi et al., 2023) and NAG, gradient-based QHD demonstrates a much more concentrated solution distribution as the iterations progress, leading to an improved global convergence. These findings motivate us to conduct a detailed investigation into gradient-based QHD and its potential in continuous optimization.

### 1.2. Overview of contributions

Our contributions are listed as follows:

- We propose **gradient-based QHD** for continuous optimization problems. With a novel Lyapunov function approach involving quantum operators, we provide a convergence analysis of gradient-based QHD in continuous time. In particular, we establish the convergence rate of gradient-based QHD in both function values (Theorem 1) and gradient norms (Theorem 4).

- We develop a quantum algorithm that simulates discrete-time gradient-based QHD to solve optimization problems (Algorithm 1). With a gate complexity linear in problem dimension $d$, this quantum algorithm is readily scalable to handle large-scale problems in practice.

- In addition to the theoretical analysis, we conduct a numerical study to evaluate the performance of gradient-based QHD in both convex and non-convex optimization. Our results show that gradient-based QHD achieves an enhanced performance compared to standard QHD and other prominent classical optimization algorithms. In some cases, gradient-based QHD

yields solutions that are an order of magnitude better than those obtained by other methods.

**Organization.** This work is structured as follows. First, we survey related classical and quantum optimization algorithms in Section 2. Next, we formulate gradient-based QHD in Section 3, with several continuous-time convergence results established in Section 4. In the subsequent Section 5, we show that gradient-based QHD can be efficiently implemented using a quantum computer. Finally, in Section 6, we present the numerical experiments comparing gradient-based QHD with several other quantum and classical optimization algorithms. We conclude this paper in Section 7.

## 2. Related work

**NAG-related algorithms and ODEs.** There has been a long history of analyzing NAG-related optimization algorithms (Giselsson & Boyd, 2014; O'donoghue & Candes, 2015). Su et al. (2016) sheds new light on the understanding and design of NAG using an ODE perspective. In (Betancourt et al., 2018; Wibisono et al., 2016; Wilson et al., 2021), a Lagrangian (or Hamiltonian) framework is used to describe a larger class of ODEs that provides a unified perspective for the acceleration phenomenon in first-order optimization. Notably, accelerated gradient descent has been investigated in non-Euclidean settings, including mirror descent (Krichene et al., 2015; Lin et al., 2019) and more generally, Riemannian manifolds (Ahn & Sra, 2020; Han et al., 2023; Kong & Tao, 2024; Siegel, 2019).

**Quantum algorithms for unconstrained optimization.** Using quantum computers to accelerate bottleneck steps in classical optimization algorithms has shown promise in achieving quantum advantage (Kerenidis & Prakash, 2020; Liu et al., 2024; Rebentrost et al., 2019). However, their practical performance requires further investigation due to the non-trivial overhead involved in extracting classical information from quantum states (Yuen, 2023). Motivated by the interplay between NAG and ODEs, another line of research proposes leveraging quantum Hamiltonian dynamics as an algorithmic surrogate for addressing unconstrained optimization problems (Leng et al., 2023a; Liu et al., 2023; Zhang et al., 2021), with recent extensions to open quantum systems (Chen et al., 2023), constrained optimization (Augustino et al., 2023), and discrete optimization (Cheng et al., 2024). This approach is particularly effective for highly non-convex problems (Leng et al., 2023b) and well-suited for hardware implementation (Kushnir et al., 2024; Leng et al., 2024). More discussions are available in Appendix B.

# 3. Gradient-based Hamiltonian dynamics

## 3.1. Classical Hamiltonian flows with gradient

Inspired by the Bregman Lagrangian (Wibisono et al., 2016) and the high-resolution ODE framework (Shi et al., 2022), we propose to study the following Lagrangian function:

$$\mathcal{L}(t, X, \dot{X}) = \frac{t^3}{2}|\dot{X}|^2 - \alpha t^3 \dot{X}^\top \nabla f(X)$$
$$- \frac{\beta t^3}{2}|\nabla f(X)|^2 - (t^3 + \gamma t^2)f(X), \quad (6)$$

where $\alpha, \beta, \gamma \in \mathbb{R}$ are real-valued parameters that will be specified later. Compared with the standard Bregman Lagrangian (1), our new Lagrangian function explicitly incorporates the gradient $\nabla f$ into the Lagrangian. This design is motivated by the convergence analysis in the high-resolution ODE, where the Lyapunov function can be interpreted as a generalized energy functional that includes gradient information. More details are provided in Appendix A.2.

By applying the Legendre transformation, we obtain the Hamiltonian function associated with (6):

$$H(t, X, P) = \sup_Y \left( P^\top Y - \mathcal{L}(t, X, Y) \right)$$
$$= \frac{1}{2}\|t^{-3/2}P + \alpha t^{3/2}\nabla f\|^2 \quad (7)$$
$$+ \frac{\beta t^3}{2}\|\nabla f(X)\|^2 + (t^3 + \gamma t^2)f(X).$$

Thus, we derive the Hamiltonian dynamics:

$$\dot{X} = \frac{\partial H}{\partial P} = \frac{1}{2t^3}P(t) + \alpha\nabla f(X(t)), \quad (8)$$

$$\dot{P} = -\frac{\partial H}{\partial X} = -\nabla^2 f(X)\left(\alpha P + (\alpha^2 + \beta)t^3\nabla f(X)\right)$$
$$- (t^3 + \gamma t^2)\nabla f(X). \quad (9)$$

**Connection with high-resolution ODEs.** It is worth noting that while our Lagrangian function shares certain similarities with high-resolution ODEs, they are not equivalent. By substituting (9) into (8), and choosing

$$\beta/\alpha = \sqrt{s}, \quad \gamma - 3\alpha = 3\sqrt{s}/2, \quad (10)$$

we can transform the Hamiltonian dynamics to a second-order ODE:

$$\ddot{X}(t) + \frac{3}{t}\dot{X}(t) + \sqrt{s}\nabla^2 f(X(t))\dot{X}$$
$$+ \left(1 + \frac{3\sqrt{s}}{2t}\right)\nabla f(X(t)) = \frac{\sqrt{s}}{2t^3}\nabla^2 f(X(t))P. \quad (11)$$

Formally, the left-hand side corresponds to the high-resolution ODE derived by Shi et al. (2022) (for details,

see Appendix A). The right-hand side of (11) is asymptotically vanishing as the momentum $P$ eventually decays to $0$.[2] Therefore, we expect the Hamiltonian dynamics to exhibit long-term behavior similar to that of high-resolution ODEs; however, we leave a detailed analysis for future research.

Due to its distinctive properties, the proposed Lagrangian function is of independent theoretical interest. In this work, we deliberately *do not* restrict the parameters $\alpha, \beta$, and $\gamma$ to the specific values associated with the high-resolution ODE case (10). This flexibility allows us to explore a broader class of dynamical systems, potentially leading to novel insights and improved algorithms for continuous optimization.

## 3.2. Canonical quantization

We introduce canonical quantization, a standard procedure that maps a classical Hamiltonian function to a quantum Hamiltonian operator. The Hamiltonian operator serves as an infinitesimal generator of a quantum evolution, which will be the core of our quantum optimization algorithms.

A classical-mechanical system is described by a Hamiltonian function $H(X, P, t)$. In contrast, a quantum-mechanical system is governed by a quantum Hamiltonian operator $\hat{H} : L^2(\mathbb{R}^d) \to L^2(\mathbb{R}^d)$. The canonical quantization procedure allows us to translate a known classical Hamiltonian function to a corresponding quantum Hamiltonian by the mapping:

$$x_j \mapsto \hat{x}_j, \quad p_j \mapsto \hat{p}_j := -i\frac{\partial}{\partial x_j}. \quad (12)$$

Here, $x_j$ and $p_j$ are the position and momentum variables describing a classical object living in a $d$-dimensional space $\mathbb{R}^d$, respectively, with the dimension indices $i \in [d]$. Correspondingly, $\hat{x}_j$ and $\hat{p}_j$ are the quantum position and momentum operators acting on wave functions $\psi(x) \in L^2(\mathbb{R}^d)$:

$$(\hat{x}_j\psi)(x) = x_j\psi(x), \quad (\hat{p}_j\psi)(x) = -i\frac{\partial}{\partial x_j}\psi(x).$$

Using this dictionary, we obtain the quantum Hamiltonian operator corresponding to the Hamiltonian function (7):

$$\hat{H}(t) = \frac{1}{2}\sum_{j=1}^d A_j^2 + \frac{\beta}{2}t^3\|\nabla f\|^2 + (t^3 + \gamma t^2)f, \quad (13)$$

where for $j = 1, \ldots, d$, the operator $A_j$ is defined by

$$A_j = t^{-3/2}\hat{p}_j + \alpha t^{3/2}\hat{v}_j, \quad \hat{v}_j\psi := \frac{\partial f}{\partial x_j}\psi. \quad (14)$$

with $\hat{v}_j$ a multiplicative operator acting on a wave function $\psi$. Due to the non-commutativity of quantum operators, the

---

[2]Details are available in Section 4.

square of the operator $A_j$ is expressed as

$$A_j^2 = t^{-3}\hat{p}_j^2 + \alpha\{\hat{p}_j, \hat{v}_j\} + \alpha^2 t^3 \hat{v}_j^2,$$

where $\{A, B\} := AB + BA$ denotes the anti-commutator of operators.

Given a quantum Hamiltonian operator $\hat{H}(t)$, the quantum evolution generated by the Hamiltonian operator is governed by the Schrödinger equation:

$$i\partial_t \Psi(t, x) = \hat{H}(t)\Psi(t, x), \qquad (15)$$

for time $0 < T_0 \leq t \leq T$, subject to an initial condition $\Psi(T_0, x) = \Psi_0(x)$. The quantum wave function $\Psi(t)$ is complex-valued, and its modulus squared $|\Psi(t)|^2$ corresponds to a probability density that characterizes the distribution of the quantum particle in the real space $\mathbb{R}^d$.

**Connection with the original QHD.** In Leng et al. (2023a), the Hamiltonian was derived from the Bregman Lagrangian via Feymann's path integral technique. Our derivation relying on canonical quantization takes a different yet complementary approach. The resulting Hamiltonian operator $\hat{H}(t)$ naturally encompasses the original QHD as a special case by choosing the parameters $\alpha = \beta = \gamma = 0$.

## 4. Convergence analysis

In this section, we focus on the convergence results of the newly derived quantum dynamics. Throughout this section, we assume $f(x^*) = 0$ and $x^* = 0$. This can always be achieved by considering the translated objective function $f(x) \leftarrow f(x + x^*) - f(x^*)$.

### 4.1. Case 1: convergence to global minimum

First, we consider a simple case where no gradient norm appears in the Hamiltonian (13), i.e., $\beta = 0$. In this case, we can prove that the dynamics converge to the global minimum of $f$.

**Theorem 1.** *Let $\beta = 0$ and $\gamma \geq \max(3\alpha, 0)$ for any $\alpha \in \mathbb{R}$. For any $1/\alpha \geq T_0 > 0$, we denote $\Psi(t, x)$ as the solution to the PDE (15) for $t \geq T_0$. Let $X_t$ be a random variable distributed according to the probability density $|\Psi(t, x)|^2$. Then, for a convex and continuously differentiable $f$, we have*

$$\mathbb{E}[f(X_t)] \leq \frac{\mathscr{K}_0 + \mathscr{D}_0}{t^2 + \omega t}, \quad \omega = \gamma - 3\alpha \geq 0,$$

*where $\mathscr{K}_0 = T_0^{-4}\langle\Psi(T_0)|(-\Delta)|\Psi(T_0)\rangle$ and*

$$\mathscr{D}_0 = \mathbb{E}\left[\|\nabla f(X_{T_0})\|^2 + 4\|X_{T_0}\|^2 + (T_0^2 + \omega T_0)f(X_{T_0})\right].$$

*In other words, $\mathbb{E}[f(X_t)] \leq O(t^{-2})$.*

**Remark 1.** $\mathscr{K}_0$ represents the initial kinetic energy (rescaled by $T_0^{-4}$). Its value is independent of $f$ and typically does not depend on the dimension $d$, e.g., when the initial state $\Psi_0$ is a standard Gaussian wave. In general, $\mathscr{D}_0$ can scale linearly in $d$ due to the presence of $\|\nabla f\|^2$.

The convergence rate is proved by constructing a Lyapunov function $\mathcal{E}(t)$ that is non-increasing in time. The Lyapunov function is defined by

$$\mathcal{E}(t) = \langle\hat{O}(t)\rangle_t := \langle\Psi(t)|\hat{O}(t)|\Psi(t)\rangle,$$

$$\hat{O}(t) = \frac{1}{2}\sum_{j=1}^d \left(t^{-2}\hat{p}_j + \alpha t\hat{v}_j + 2\hat{x}_j\right)^2 + \left(t^2 + \omega t\right)f.$$

Here, $\omega = \gamma - 3\alpha \geq 0$ because $\gamma \geq \max(3\alpha, 0)$.

**Lemma 2.** *Let $\beta = 0$ and $\gamma \geq \max(3\alpha, 0)$. For any $t > 0$, we have $\dot{\mathcal{E}}(t) \leq 0$.*

In Lemma 2, we prove that the function $\mathcal{E}(t)$ is non-increasing in time, as a result,

$$t^2\langle f\rangle_t \leq \mathcal{E}(t) \leq \mathcal{E}(T_0) \implies \langle f\rangle_t \leq \frac{\mathcal{E}(T_0)}{t^2}.$$

Moreover, we note that

$$\mathcal{E}(T_0) \leq \langle\Psi(t)\left|\frac{1}{T_0^4}(-\Delta) + \alpha^2 T_0^2\|\nabla f\|^2 + 4\|x\|^2\right|\Psi(t)\rangle$$
$$+ (T_0^2 + \omega T_0)\langle\Psi(t)|f|\Psi(t)\rangle,$$

which proves Theorem 1.

The details of Lemma 2 is presented in Appendix C.2. The technical proof heavily relies on the commutation relations between various non-commuting quantum operators that appeared in the Lyapunov function. We summarize the common commutation relations used in this work in Lemma 3, which might be of independent interest in future work.

**Lemma 3** (Commutation relations). *Let $A_j$, $p_j$, and $x_j$ be the same as above. For any $1 \leq j, k \leq d$, we have the following identities:*

1. $i[A_j^2, f] = t^{-3}\{p_j, v_j\} + 2\alpha v_j^2,$

2. $i[f, \{A_j, x_j\}] = -2t^{-3/2}x_j v_j,$

3. $i[A_j^2, x_k^2] = \begin{cases} \frac{2}{t^3}\{p_j, x_j\} + 4\alpha x_j v_j & (j = k) \\ 0 & (j \neq k) \end{cases},$

4. $i[A_j^2, \{A_k, x_k\}] = \begin{cases} \frac{4}{t^{3/2}}A_j^2 & (j = k) \\ 0 & (j \neq k) \end{cases},$

5. $i[v_j^2, \{A_k, x_k\}] = -4t^{-3/2}\left(\frac{\partial^2 f}{\partial x_j x_k}\right)x_k v_j.$

For the proof, please refer to Appendix C.1.

## 4.2. Case 2: convergence to first-order stationary point

We denote the function $G(x)$ as the square of the gradient norm of $f$, i.e.,

$$G(x) := \nabla f(x)^\top \nabla f(x) = \sum_{j=1}^{d} \left| \frac{\partial f(x)}{\partial x_j} \right|^2. \quad (16)$$

**Theorem 4.** *Let $\gamma \geq \max(3\alpha, 0)$ and $\beta > 0$. For any $\min(1/\alpha, \sqrt{2/\beta}) \geq T_0 > 0$, we denote $\Psi(t, x)$ as the solution to the PDE (15). Let $X_t$ be a random variable distributed according to the probability density $|\Psi(t, x)|^2$. Then, for a convex and continuously differentiable $f$ such that its gradient norm satisfies the following identity:*

$$G(x) - \nabla G(x)^\top x \leq 0, \quad (17)$$

*we have*

$$\mathbb{E}[\|\nabla f(X_t)\|^2] \leq \frac{2(\mathcal{K}_0 + \mathcal{D}_0')}{\beta t^2},$$

*where $\mathcal{K}_0$ is the same as in Theorem 1 and*

$$\mathcal{D}_0' = \mathbb{E}\left[ 2\|\nabla f(X_{T_0})\|^2 + 4\|X_{T_0}\|^2 + (T_0^2 + \omega T_0) f(X_{T_0}) \right],$$

*where $\omega = \gamma - 3\alpha$. In other words, $\mathbb{E}[\|\nabla f(X_t)\|^2] \leq O(t^{-2})$.*

**Remark 2.** A sufficient condition for the identity (17) is that $G(x)$ is convex. In this case, the global minimizer of $G(x)$ must be $x^*$ and (17) holds. However, this does not always require the objective function $f$ to be convex. For example, consider $f(x) = \sqrt{x}$ for $x > 0$. While $f$ is a concave function, $G(x) = (f')^2 = \frac{1}{4x}$ is a convex function for $x > 0$.

Similarly, the proof of Theorem 4 exploits a Lyapunov function approach. We define

$$\mathcal{F}(t) = \langle \hat{J}(t) \rangle_t := \langle \Psi(t)|\hat{J}(t)|\Psi(t) \rangle,$$

$$\hat{J}(t) = \frac{1}{2} \sum_{j=1}^{d} \left( t^{-2} \hat{p}_j + \alpha t \hat{v}_j + 2\hat{x}_j \right)^2 + \frac{\beta}{2} t^2 G + (t^2 + \omega t) f,$$

with $\omega = \gamma - 3\alpha \geq 0$. Due to the positivity of $(t^{-2}p_j + \alpha t v_j + 2x_j)^2$ and $f$, we have

$$\langle \|\nabla f\|^2 \rangle_t \leq \frac{2}{\beta t^2} \mathcal{F}(t) \leq \frac{2}{\beta t^2} \mathcal{F}(0),$$

where the last step follows from Lemma 5. This proves Theorem 4.

**Lemma 5.** *Let $\gamma > 0$ and $\alpha \geq \max(\beta, 0)$. If (17) holds, we have $\dot{\mathcal{F}}(t) \leq 0$ for any $t > 0$.*

The proof is left in Appendix C.3.

## 5. Quantum algorithms and complexity analysis

In this section, we study the time discretization of gradient-based QHD, which facilitates the simulation of the quantum dynamics in a (fault-tolerant) quantum computer.

### 5.1. Time discretization of the quantum Hamiltonian dynamics

Recall that the gradient-based QHD dynamics are governed by the differential equation (15). Let $U(t)$ be the time-evolution operator that maps an initial state $|\Psi_0\rangle$ to the solution state $|\Psi(t)\rangle$ at time $t \in [0, T]$, i.e.,

$$U(t)\Psi(0) = \Psi(t) \quad \forall t \in [0, T].$$

Formally, the time-evolution operator can be obtained by a sequence of infinitesimal time evolution of the quantum Hamiltonian $\hat{H}(t)$:

$$U(t) = \lim_{K \to \infty} e^{-ih\hat{H}(t_K)} e^{-ih\hat{H}(t_{K-1})} \dots e^{-ih\hat{H}(t_1)},$$

where $K$ is a positive integer, $h = t/K$ and $t_k = kh$ for $1 \leq k \leq N$. Note that the gradient-based QHD Hamiltonian can be decomposed in the form $\hat{H}(t_k) = H_{k,1} + H_{k,2} + H_{k,3}$, where

$$H_{k,1} = -\frac{1}{2t_k^3}\Delta, \quad H_{k,2} = \frac{\alpha}{2}\{-i\nabla, \nabla f\},$$

$$H_{k,3} = \frac{(\alpha^2 + \beta)}{2} t^3 \|\nabla f\|^2 + (t^3 + \gamma t^2) f.$$

Therefore, we can further decompose a short-time evolution step using the product formula (i.e., operator splitting):

$$e^{-ih\hat{H}(t_k)} \approx e^{-ihH_{k,1}} e^{-ihH_{k,2}} e^{-ihH_{k,3}}. \quad (18)$$

Since all the Hamiltonians $H_{k,1}$, $H_{k,2}$, and $H_{k,3}$ can be efficiently simulated using a quantum computer, we obtain a quantum algorithm that implements gradient-based QHD to solve large-scale optimization problems, as summarized in Algorithm 1.

**On the choice of step size $h$.** It is shown in Childs et al. (2021) that the product formula will introduce a "simulation error" such that

$$\left\| e^{-ih\hat{H}(t_k)} - e^{-ihH_{k,1}} e^{-ihH_{k,2}} e^{-ihH_{k,3}} \right\|$$

$$\leq \frac{h^2}{2} \sum_{1 \leq i \neq j \leq 3} \|[H_{k,i}, H_{k,j}]\|.$$

A formal calculation shows that the commutator norm scales as $\mathcal{O}(t_k^3)$, which suggests $h \sim t_k^{-3/2}$ may be needed to control the simulation error in each time step. However, in

---

**Algorithm 1** Gradient-based QHD with fixed step size

**Classical Input:** Hamiltonian parameters $\alpha, \beta, \gamma$, step size $h$, number of iterations $K$.
**Quantum Input:** an initial guess state $|\Psi_0\rangle$
**Output:** a classical point $\xi \in \mathbb{R}^d$.

---

Initialize the quantum register to $|\Psi_0\rangle$.
**for** $k = 1$ **to** $K$ **do**
    Determine $t_k = kh$.
    Implement a quantum circuit $U_k$ as described in (18).
    Compute $|\Psi_k\rangle = U_k |\Psi_{k-1}\rangle$.
**end for**
Measure the final quantum state $|\Psi_K\rangle$ with the position observable $\hat{x}$ to obtain a sample point $\xi \in \mathbb{R}^d$.

---

the numerical experiments, we observe that a much larger step size $h$ can still result in the convergence of the discrete-time gradient-based QHD. This observation aligns with our experience with the NAG method, where convergence is achieved with a step size proportional to $1/L$, irrespective of the continuous-time dynamics. As a result, we treat the step size $h$ as an independent parameter in the complexity analysis. A complete understanding of the convergence of the discrete-time algorithm, however, is left for future study.

**Remark 3.** Quantum simulations of time-dependent Hamiltonians constitute an active research area, with a growing body of literature addressing this topic (e.g., (An et al., 2021; 2022; Berry et al., 2020; Childs et al., 2022; Mizuta et al., 2024)). These developments pave the way for more advanced implementations of gradient-based QHD, potentially offering improved asymptotic complexity. A detailed exploration of such implementations is left for future work.

### 5.2. Complexity analysis

Now, we analyze the computational cost of Algorithm 1. In our analysis, we assume the quantum computer has access to the function $f$ and its gradient via the following quantum circuits:

$$O_f \colon |x\rangle |z\rangle \mapsto |x\rangle |f(x) + z\rangle ,$$
$$O_{\nabla f} \colon |x\rangle |z\rangle \mapsto |x\rangle |\nabla f(x) + z\rangle .$$

The quantum circuits $O_f$ and $O_{\nabla f}$ are often called quantum *zeroth*- and *first-order* oracles. They can be efficiently constructed by quantum arithmetic circuits when the expressions of $f$ and $\nabla f$ are known.

**Remark 4.** The requirement for a quantum first-order oracle $O_{\nabla f}$ can potentially be eliminated by leveraging Jordan's algorithm (Jordan, 2005), which estimates gradients using only a zeroth-order oracle $O_f$. However, without astrong smoothness assumptions on the objective function $f$, the query complexity of obtaining an $\epsilon$-approximate gradient

typically scales as $\mathcal{O}(\sqrt{d}/\epsilon)$ (Gilyén et al., 2019a). In this work, we focus on the convergence properties of gradient-based QHD, leaving the incorporation of quantum gradient estimation techniques for future research.

A crucial step in Algorithm 1 is to implement the quantum unitary operator $U_k$ based on the operator splitting formula (18). We note that the sub-Hamiltonians $H_{k,1}$ and $H_{k,3}$ are fast-forwardable, and the operator $H_{k,2}$ can be simulated by invoking Quantum Singular Value Transformation (QSVT). Combining these technical results together, we end up with the overall complexity of the quantum algorithm, as summarized in Theorem 6.

**Theorem 6.** *Let $f$ be L-Lipschitz and $|\Psi_0\rangle$ be a sufficiently smooth function. Then, we can implement Algorithm 1 for $K$ iterations using $\mathcal{O}(K)$ queries to the quantum zeroth-order oracle $O_f$ and $\widetilde{\mathcal{O}}(\alpha dh KL)$ queries to the quantum first-order oracle $O_{\nabla f}$ and its inverses.*[3]

The details proof of Theorem 6, including the efficient simulation of $H_{k,2}$ via QSVT, is presented in Appendix D.

## 6. Numerical experiments

In this section, we conduct extensive numerical experiments to evaluate the performance of gradient-based QHD and compare it with other prominent optimization algorithms.

### 6.1. Experiment setup and implementation details

Let $f \colon \mathbb{R}^d \to \mathbb{R}$ be an objective function with gradient $\nabla f(x)$. Given an optimization algorithm initialized with a random sample drawn from a fixed distribution $\rho_0$, the algorithm's output after $k$ iterations can be represented by a random variable $X_k \in \mathbb{R}^d$. We denote $\mathbb{E}[f(X_k)]$ as the expectation value of the objective function and $\mathbb{E}[\|\nabla f(X_k)\|^2]$ as expected gradient norm at iteration $k$. To assess the algorithm's performance, we define the success probability after $k$ iterations as

$$\mathbf{P}_k \coloneqq \mathbb{P}[f(X_k) - f(x^*) \le \delta].$$

where $\delta > 0$ is a pre-defined error threshold. For all the subsequent experiments, we set $\delta = 1$.

We remark that the iteration steps in gradient-based QHD (as shown in Algorithm 1) are more intricate than those in classical methods such as SGDM and NAG. As demonstrated in the proof of Lemma 9, the query and gate complexity per iteration of gradient-based QHD scales as $\widetilde{O}(d)$. In contrast, each iteration of SGDM/NAG involves only a single query to $\nabla f$, with a time complexity of $\mathcal{O}(d)$. Therefore, in terms

---

[3]Here, the $\widetilde{\mathcal{O}}$ notation suppresses poly-logarithmic factors in the error parameter $\epsilon$. The parameter $\epsilon > 0$ represents the error budget in the Hamiltonian simulation, as detailed in Lemma 9.

of overall runtime, gradient-based QHD remains asymptotically comparable to NAG, which justifies our comparison based on the iteration count.

To evaluate the classical methods such as SGDM and NAG, we estimate the expectation values and success probabilities using a sample of 1000 independent runs. Each run begins with a uniformly random initial guess and proceeds for $k$ iterations. For the quantum methods, since the probability density function can be explicitly derived from the quantum state vector, expectation values and success probabilities are computed via numerical integration.

The numerical simulations of the quantum algorithms, including QHD and gradient-based QHD, are performed on a MacBook equipped with an M4 chip. Additional details on the numerical methods employed are provided Appendix E.1.

## 6.2. Convex optimization

To evaluate performance, we conduct a numerical comparison of gradient-based QHD against three baseline algorithms, including SGDM, NAG, and QHD, for convex optimization. The test function used is

$$f(x, y) = \frac{(x+y)^4}{256} + \frac{(x-y)^4}{128},\tag{19}$$

which is a convex yet non-strongly convex function, with a singular Hessian at its unique minimum $(0, 0)$. Notably, the gradient of this function does not satisfy the Lipschitz continuity condition. This flat geometry presents significant challenges for classical methods that rely heavily on curvature information, making it a suitable benchmark for comparative evaluation. All methods are executed with a fixed step size $h = 0.2$.[4] For the quantum variants, the initial evolution time is set to $t_0 = 1$. The parameters of gradient-based QHD are configured as $\alpha = -0.1$, $\beta = 0$, and $\gamma = 5$.

The performance of these optimization algorithms is visualized in Figure 3, where two key metrics are employed to access convergence: the average function values $\mathbb{E}[f(X_k)]$ (depicted in the left subplot) and the average gradient norm $\mathbb{E}[\|\nabla f(X_k)\|^2]$ (depicted in the right subplot). Both quantities are tracked over iterations $1 \leq k \leq 25$. The results reveal distinct convergence behaviors. While the (classical) QHD exhibits a slower convergence rate compared to NAG, the gradient-based QHD stands out by achieving a remarkably faster convergence rate, outperforming all other algorithms. This superior performance highlights the effectiveness of incorporating gradient-based techniques into QHD, particularly for challenging optimization landscapes.

---

[4]We have tested various step sizes ($h \in [0.05, 0.5]$) for gradient-based QHD and observed similar convergence behavior. To maintain consistency, we fix $h = 0.2$ in all comparisons.

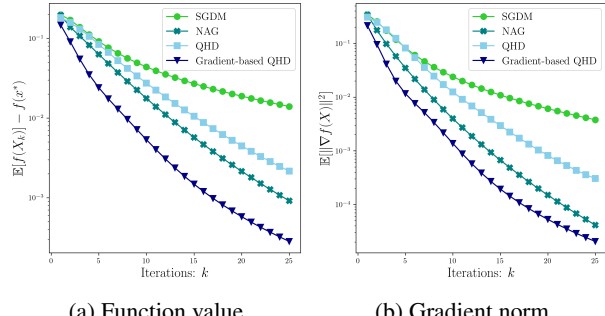

(a) Function value      (b) Gradient norm

*Figure 3.* Numerical comparison of various optimization algorithms on the convex objective function (19), including function values and success probability.

## 6.3. Non-convex optimization

More details on these test problems are available in Appendix E.2.

We now turn our attention to the numerical comparison of gradient-based QHD against three baseline algorithms, including SGDM, NAG, and QHD, in non-convex optimization settings. Non-convexity introduces significant challenges for classical first-order methods, as local gradient information alone is often insufficient to distinguish the global minimum from other spurious local optima.

To illustrate these challenges, we evaluate a variety of non-convex optimization problem instances characterized by diverse landscape features:

(i) **Michalewicz function** (Figure 4): This function features a flat plateau and a unique global minimum hidden within a sharp valley, posing a difficult search problem.

(ii) **Cube-Wave function** (Figure 5): With over ten local minima (four of which are global minima) concentrated within the cube $[-2, 2]^2$, this function exemplifies a rugged landscape.

(iii) **Rastrigin function** (Figure 6): This function presents a highly oscillatory landscape with a global minimum at the origin, making it notoriously challenging for optimization algorithms.

Due to these intricate characteristics, all three problems are recognized as particularly difficult for classical first-order methods. Additional details about these test functions are provided in Appendix E.2.

For the two quantum algorithms, the evolution time starts from $t_0 = 0$. In gradient-based QHD, the parameters are set to $\alpha = -0.05$, $\beta = 0$, and $\gamma = 5$.

Despite the diversity of non-convex test problems, gradient-based QHD consistently delivers robust and favorable performance. Compared to both QHD and the classical algorithms, it achieves a significantly faster convergence rate

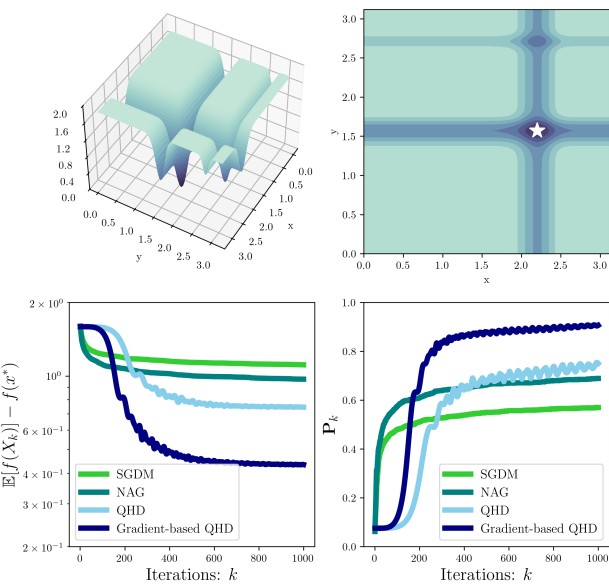

*Figure 4.* Numerical comparison of various optimization algorithms on the Michalewicz function, including function values and success probability.

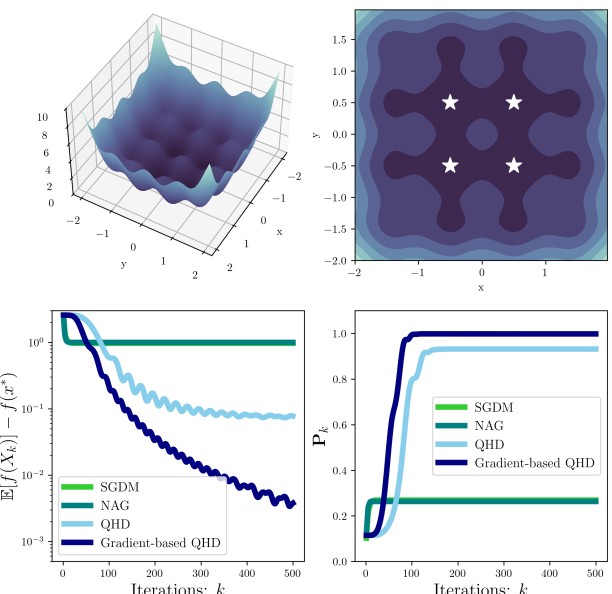

*Figure 5.* Numerical comparison of various optimization algorithms on the Cube-Wave function, including function values and success probability.

and yields notably lower terminal objective function values. In the Cube-Wave function, for instance, the final objective value obtained by gradient-based QHD is nearly an order of magnitude lower than that of QHD and two orders of magnitude lower than those achieved by SGDM and NAG.

Further numerical analysis highlights that gradient-based QHD attains a higher success probability across all problem instances, indicating that its final states are tightly concentrated around the global minimizer. In summary, by leveraging gradient information within the quantum Hamiltonian

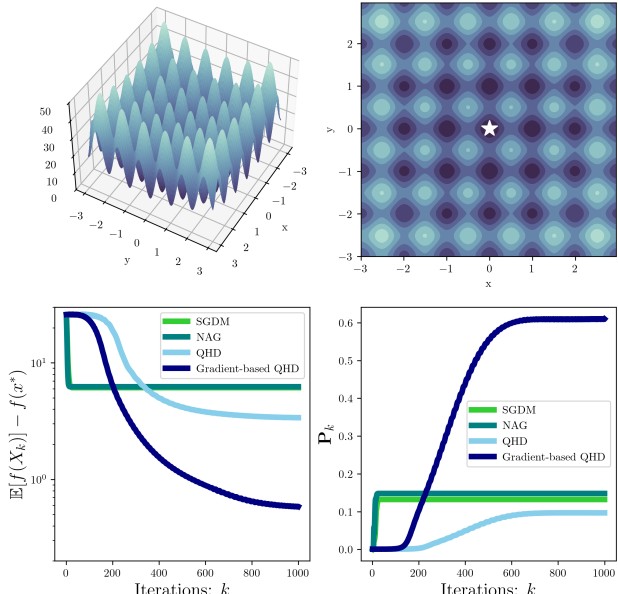

*Figure 6.* Numerical comparison of various optimization algorithms on the Rastrigin function, including function values and success probability.

framework, gradient-based QHD demonstrates enhanced *global* convergence properties, outperforming QHD and classical optimization methods.

## 7. Conclusion and Future Work

In this paper, we propose gradient-based QHD for continuous optimization problems without constraints. We prove the convergence of the gradient-based QHD dynamics in both function values and gradient norms via a Lyapunov function approach. We also discuss an efficient implementation of discrete-time gradient-based QHD using a fault-tolerant quantum computer. Our numerical results show that gradient-based QHD achieves improved convergence with a higher chance of identifying the global minimum in a sophisticated optimization landscape.

Our theoretical analysis has primarily focused on the convergence of gradient-based QHD in continuous time, while the long-term behavior of the discrete-time algorithm deserves further investigation. The numerical experiments are limited to 2D problems due to the exponential growth of computational overhead. Developing new numerical techniques could help evaluate the advantages of quantum Hamiltonian-based algorithms for high-dimensional optimization.

## Software and Data

The source code of the experiments is available at https://github.com/jiaqileng/Gradient-Based-QHD.

## Acknowledgements

J. L. is partially supported by the Simons Quantum Post-doctoral Fellowship and a Simons Investigator Award in Mathematics through Grant No. 825053. B. S. is partially supported by a startup fund from SIMIS and Grant No.12241105 from NSFC. Most of this work was completed at the University of California, Berkeley, during B.S.'s visit in the fall of 2024.

## Impact Statement

This paper presents work whose goal is to advance the field of Machine Learning. There are many potential societal consequences of our work, none of which we feel must be specifically highlighted here.

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

# A. Review of accelerated gradient descent

## A.1. Accelerated gradient descent as differential equations

Accelerated gradient descent methods are fundamental in both theory and practice. Nesterov (Nesterov, 1983) proposed the first accelerated gradient method that has the following update rules (where $s > 0$ is the step size):

$$x_k = y_{k-1} - s\nabla f(y_{k-1}), \tag{20a}$$

$$y_k = x_k + \frac{k-1}{k+2}(x_k - x_{k-1}), \tag{20b}$$

It is known that Nesterov's gradient descent achieves the optimal convergence rate among all gradient-based methods.

On the other hand, there has been a long-lasting research attempting to relate gradient-based optimization algorithms with differential equations. A seminal work by Su et al. (Shi et al., 2022) proposed a second-order differential equation to capture the acceleration phenomenon in Nesterov's algorithm. For sufficiently small step size $s$, the continuous-time limit of (20) is given by the following ordinary differential equation,

$$\ddot{X} + \frac{3}{t}\dot{X} + \nabla f(X) = 0, \tag{21}$$

for $t > 0$, with initial conditions $X(0) = x_0$ and $\dot{X}(0) = 0$. The convergence rate of the ODE is $O(t^{-2})$, which matches that of the discrete-time algorithm (20).

The ODE framework of accelerated gradient descent was later generalized via a variational formulation of the underlying dynamics. Wibisono, Wilson, and Jordan (Wibisono et al., 2016) proposed to consider the Bregman Lagrangian,

$$\mathcal{L}(X, V, t) = e^{\alpha_t + \gamma_t}\left(\frac{1}{2}\left|e^{-\alpha_t}V\right|^2 - e^{\beta_t}f(X)\right), \tag{22}$$

where $t \geq 0$ is the time, $X \in \mathbb{R}^d$ is the state vector, and $V \in \mathbb{R}^d$ is the velocity.[5] Given the Lagrangian function $\mathcal{L}(X, V, t)$, we can consider the following variational problem:

$$\min_{X_t} J(X_t) = \int_0^\infty \mathcal{L}(X, \dot{X}_t, t)dt, \tag{23}$$

where $J(X_t)$ is a functional defined on smooth curves $\{X_t : t \in [0, \infty)\}$. From the calculus of variations, a curve that minimizes the functional $J(X_t)$ necessarily satisfies the Euler-Lagrange equation:

$$\frac{d}{dt}\left(\frac{\partial \mathcal{L}}{\partial V}(X_t, \dot{X}_t, t)\right) = \frac{\partial \mathcal{L}}{\partial X}(X_t, \dot{X}_t, t). \tag{24}$$

Specifically, if we choose $\alpha_t = -\log(t)$, $\beta_t = \gamma_t = 2\log(t)$, the resulting Euler-Langrage equation is exactly the continuous-time limit of Nesterov's accelerated gradient descent (21). It is also shown that, if $\alpha_t$, $\beta_t$, and $\gamma_t$ satisfies the following ideal scaling conditions,

$$\dot{\beta}_t \leq e^{\alpha_t}, \quad \dot{\gamma}_t = e^{\alpha_t}, \tag{25}$$

the solutions to the Euler-Lagrange equation satisfy

$$f(X_t) - f(x^*) \leq O(e^{-\beta_t}), \tag{26}$$

which gives a convergence rate of the dynamical system in continuous time.

---

[5]Here, we give a simplified version of the Bregman Lagrangian in which the Bregman divergence is given by the standard Euclidean distance; for details, see (Wibisono et al., 2016).

## A.2. Understanding acceleration via high-resolution ODEs

While the continuous-time formulations of accelerated gradient descent provide a more transparent perspective on the acceleration phenomenon and allow us to introduce the rich toolbox from ODE theory, they offer little understanding of different accelerated gradient descent algorithms with the same continuous-time limit. For example, Polyak's heavy-ball method and NAG have the same continuous-time limit, however, they exhibit strikingly different behaviors in practice: the heavy-ball method generally only achieves local acceleration, while NAG is an acceleration method applicable to all initial values of the iterate (Lessard et al., 2016).

The difference between the two algorithms lies in a gradient correction step that only exists in NAG. Inspired by the dimensional-analysis strategies in fluid mechanics, Shi et al. (2022) developed a high-resolution ODE framework to reflect the gradient correction effect in different algorithms with the same low-resolution continuous-time limit. The high-resolution ODEs for NAG are as follows.

$$\ddot{X}(t) + \frac{3}{t}\dot{X}(t) + \sqrt{s}\nabla^2 f(X(t))\dot{X}(t) + \left(1 + \frac{3\sqrt{s}}{2t}\right)\nabla f(X(t)) = 0, \tag{27}$$

for $t \geq 3\sqrt{s}/2$, with $X(3\sqrt{s}/2) = x_0$, $\dot{X}(3\sqrt{s}/2) = -\sqrt{s}\nabla f(x_0)$.

In contrast, the high-resolution ODE for the heavy-ball method does not have the higher-order correction term $\sqrt{s}\nabla^2 f(X(t))\dot{X}(t)$, which explains how the gradient correction step improves the overall convergence performance of NAG over the heavy-ball method. The high-resolution ODE framework also motivates the design of a new family of accelerated gradient descent algorithms that maintain the convergence rate of NAG.

To prove the convergence of the high-resolution ODEs, Shi et al. (2022) employs the following Lyapunov function (see Shi et al. (2022, Eq. (4.36))):

$$\mathcal{E}(t) = t\left(t + \frac{\sqrt{s}}{2}\right)(f(X) - f(x^*)) + \frac{1}{2}\|t\dot{X} + 2(X - x^*) + t\sqrt{s}\nabla f(X)\|^2. \tag{28}$$

Let $X(t)$ be the solution to (27), it is proven in Shi et al. (2022, Lemma 5) that for all $t \geq 3\sqrt{s}/2$

$$\frac{\mathrm{d}\mathcal{E}(t)}{\mathrm{d}t} \leq -\left[\sqrt{s}t^2 + \left(\frac{1}{L} + \frac{s}{2}\right)t + \frac{\sqrt{s}}{2L}\right]\|\nabla f(X)\|^2 < 0. \tag{29}$$

As a direct consequence, for any $t \geq 3\sqrt{s}/2$, we have

$$f(X(t)) - f(x^*) \leq \frac{(4 + 3sL)\|x_0 - x^*\|^2}{t(2t + \sqrt{s})}. \tag{30}$$

# B. Two paradigms of quantum optimization

Based on how the solution is encoded in a quantum state, there are two major paradigms in designing quantum algorithms for continuous optimization problems. In this section, we briefly discuss the two paradigms and compare their respective pros and cons.

**Solution vector as a quantum state.** The first paradigm uses *amplitude encoding*, where an $n$-dimensional vector $\mathbf{v}$ is encoded into a $q$-qubit state with $q = \lceil \log_2(n) \rceil$:

$$|\mathbf{v}\rangle = \frac{1}{\|\mathbf{v}\|}\sum_{j=1}^{n} \mathbf{v}_j |j\rangle,$$

where $\{|j\rangle\}_j^{2^q-1}$ is the set of computational basis. This approach encompasses a vast majority of works in quantum optimization, including (Kerenidis & Prakash, 2020; Liu et al., 2024; Rebentrost et al., 2019). In this encoding scheme, the solution vector can be represented using $\mathcal{O}(\log(n))$ qubits and it allows us to exploit the rich quantum numerical linear algebra toolbox to accelerate existing classical algorithms. Nevertheless, the downside is that the recovery of the classical vector $\mathbf{v}$ from its amplitude-encoded state $|\mathbf{v}\rangle$, a task known as *quantum state tomography*, would inevitably incur a $\Theta(n/\epsilon)$ overhead due to the Heisenberg limit, where $\epsilon$ is the readout precision (Yuen, 2023). Therefore, the quantum state tomography can nullify the potential exponential quantum speedup in the computation.

**Superposition of all possible solutions.** Another paradigm uses *basis encoding*, where an $n$-dimensional vector $\mathbf{v}$ corresponds to a unique computational basis $|b_v\rangle$. To see how this works, we assume that each element in the real-valued vector $\mathbf{v}$ is represented by a fixed-point number $v_j$ with bit length $q$. Therefore, we can uniquely enumerate all possible solutions (corresponding to all possible fixed-point numbers) in the $n$-dimensional space using $(2^q)^n$ computational basis, or equivalently, $nq$ qubits. This encoding scheme is similar to how modern computers store an array with fixed/floating-point arrays. Nevertheless, the difference is that quantum computers can produce a *superposition* of basis states, i.e.,

$$|\Psi\rangle = \sum_{\mathbf{x}} \sqrt{\rho(\mathbf{x})} \, |\mathbf{x}\rangle \,,$$

where $\rho$ is a probability distribution over the whole space. In this case, measuring the quantum state $|\Psi\rangle$ is equivalent to sampling a point from the distribution $\rho$. Solving an optimization problem amounts to preparing an approximation of the Dirac-delta distribution at the minimizer $x^*$, i.e., the state $|\mathbf{x}^*\rangle$. Compared to the first paradigm, there are two major advantages of this approach: First, there is no obvious fundamental limitation on extracting information from the quantum register, as we can prepare a Dirac-delta-like state for which the probability of obtaining a fixed solution can be arbitrarily close to 1. Second, the superposition of solutions $|\Psi\rangle$ is a natural quantum wave function, so we can design a solution path by exploiting the toolbox of continuous-space quantum mechanics, which is historically less explored in the quantum computation literature. The drawback, however, is that we will not have exponentially improved space/qubit complexity to represent the solution. The first approach in principle only uses $\mathcal{O}(\log_2(n))$ qubits to represent a solution vector, while representing the superposition state $|\Psi\rangle$ requires $\mathcal{O}(n)$ qubits for an $n$-dimensional vector.

## C. Technical details of convergence analysis

### C.1. Commutation relations in gradient-based QHD

This section proves the commutation relations in Lemma 3.

**Lemma 7.** *Let $g \colon \mathbb{R}^d \to \mathbb{R}$ be a smooth function. We have*

$$i[\hat{p}_j^2, g] = \{\hat{p}_j, \partial_j g\}.$$

*Proof.* Let $\varphi$ be a test function. Note that

$$(\hat{p}_j^2 g)(\varphi) = -\frac{\partial^2}{\partial x_j^2}(g\varphi) = -(\partial_{jj}g\varphi + 2\partial_j g\partial_j\varphi + g\partial_{jj}\varphi), \quad (g\hat{p}_j^2)(\varphi) = g\left(-\frac{\partial^2}{\partial x_j^2}\varphi\right) = -g\partial_{jj}\varphi.$$

Therefore,

$$i[\hat{p}_j^2, g]\varphi = -i\left(\partial_{jj}g\varphi + 2\partial_j g\partial_j\varphi\right).$$

Meanwhile, we find that

$$\{\hat{p}_j, \partial_j g\}\varphi = (-i\partial_j)(\partial_j g\varphi) + \partial_j g\left(-i\partial_j\varphi\right) = -i\left(\partial_{jj}g\varphi + 2\partial_j g\partial_j\varphi\right),$$

which concludes the proof. $\square$

**Lemma 8.** *Let $g \colon \mathbb{R}^d \to \mathbb{R}$, $h \colon \mathbb{R}^d \to \mathbb{R}$ be two smooth functions. We have*

$$i[\{\hat{p}_j, h\}, g] = 2h(\partial_j g).$$

*Proof.* Let $\varphi$ be a test function. Direct calculation shows that

$$\begin{aligned}
i[\{\hat{p}_j, h\}, g]\varphi &= i\left[(p_j h + hp_j)(g\varphi) - g(p_j h + hp_j)\varphi\right] \\
&= i\left[p_j(hg\varphi) + h(p_j g)\varphi - gp_j(h\varphi) - gh(p_j\varphi)\right] \\
&= 2ih(p_j g)\varphi = 2h(\partial_j g)\varphi,
\end{aligned}$$

which implies $i[\{\hat{p}_j, h\}, g] = 2h(\partial_j g)$. This operator is again a multiplicative operator that commutes with both $g$ and $h$. $\square$

Now, we are ready to prove Lemma 3.

*Proof.* 1. Recall that

$$A_j = t^{-3/2}p_j + \alpha t^{3/2}v_j, \quad v_j = \frac{\partial f}{\partial x_j}, \quad A_j^2 = t^{-3}p_j^2 + \alpha\{p_j, v_j\} + \alpha^2 t^3 v_j^2.$$

Therefore,

$$i[A_j^2, f] = i[t^{-3}p_j^2 + \alpha\{p_j, v_j\}, f] = t^{-3}\{p_j, v_j\} + 2\alpha v_j^2.$$

The last identity invokes Lemma 7 and Lemma 8.

2. Since the $v_j$ part in $A_j$ commutates with $x$ and $f$, we can drop it from the commutator:

$$i[f, \{A_j, x_j\}] = i[f, \{t^{-3/2}p_j, x_j\}] = -it^{-3/2}[\{p_j, x_j\}, f] = -t^{-3/2}x_j v_j,$$

where we use Lemma 8 in the last step.

3. By dropping the $v_j^2$ part in $A_j^2$, we get

$$i[A_j^2, x_k^2] = i[t^{-3}p_j^2 + \alpha\{p_j, v_j\}, x_k^2] = t^{-3}i[p_j^2, x_k^2] + \alpha[\{p_j, v_j\}, x_k^2].$$

By Lemma 7 and Lemma 8, we obtain the following:

$$i[A_j^2, x_k^2] = \begin{cases} 0 & (j \neq k) \\ 2t^{-3}\{p_j, x_j\} + 4\alpha x_j v_j & (j = k). \end{cases}$$

4. It can be readily verified that $[A_j, A_k] = 0$ and $[A_j, x_k] = 0$ for any $j \neq k$. Therefore, if $j \neq k$, we will have

$$i[A_j^2, \{A_k, x_k\}] = 0.$$

When $j = k$, we first observe that

$$i[A_j, x_j] = i[t^{-3/2}p_j, x_j] = t^{-3/2}i[p_j, x_j] = t^{-3/2}$$

due to the canonical commutation relation $i[p_j, x_j] = 1$. By leveraging the commutation relation between $A_j$ and $x_j$,

$$\begin{aligned} i[A_j^2, \{A_j, x_j\}] &= i\left(A_j^3 x_j + A_j^2 x_j A_j - A_j x_j A_j^2 - x_j A_j^3\right) \\ &= i\left(A_j^2(x_j A_j - it^{-3/2}) + A_j^2 x_j A_j - A_j x_j A_j^2 - (A_j x_j + it^{-3/2})A_j^2\right) \\ &= i\left(2A_j(x_j A_j - it^{-3/2})A_j - 2A_j x_j A_j^2 - 2it^{-3/2}\right) \\ &= 4t^{-3/2}A_j^2. \end{aligned}$$

5. This commutation relation is a direct consequence of Lemma 8. By dropping the $v_k$ part in $A_k$, we have

$$i[v_j^2, \{A_k, x_k\}] = -it^{-3/2}[\{p_j, x_k\}, v_j^2] = -4t^{-3/2}x_k v_j(\partial_k v_j) = -4t^{-3/2}\left(\frac{\partial^2 f}{\partial x_j x_k}\right)x_k v_j.$$

□

## C.2. Proof of Lemma 2

*Proof.* By the definition of the Lyapunov function, we have

$$\frac{\mathrm{d}}{\mathrm{d}t}\mathcal{E}(t) = \langle \partial_t \hat{O}(t) + i[\hat{H}(t), \hat{O}(t)] \rangle_t, \tag{31}$$

where $[A, B] := AB - BA$ denotes the commutator of operators. In the following calculation, we omit the hat over quantum operators to simplify the notation.

First, we calculate the $\partial_t O(t)$ part. Direct calculations yield that

$$\partial_t O(t) = \sum_{j=1}^{d} \left( -\frac{2}{t^5}p_j^2 + \alpha^2 t v_j^2 + 2\alpha x_j v_j - \frac{\alpha}{2t^2}\{p_j, v_j\} - \frac{2}{t^3}\{p_j, x_j\} \right) + (2t + \omega)f. \tag{32}$$

As for the commutator part, it is worth noting that

$$
\begin{aligned}
O(t) &= \frac{1}{2}\sum_{j=1}^{d}(t^{-1/2}A_j + 2\hat{x}_j)^2 + \left(t^2 + \omega t\right)f \\
&= \frac{1}{t}H(t) + \sum_{j=1}^{d}\left(2x_j^2 + \frac{1}{t^{1/2}}\{A_j, x_j\}\right) - 3\alpha t f.
\end{aligned}
\tag{33}
$$

Therefore, we have

$$i[H(t), O(t)] = i\left[H(t), \sum_{j=1}^{d}\left(2x_j^2 + \frac{\{A_j, x_j\}}{t^2}\right) - 3\alpha t f\right] \tag{34}$$

Invoking the commutation relations 1-4 in Lemma 3 to simplify (34) and combining it with (32), we obtain the following identity:

$$\partial_t O + i[H, O] = (2t + \omega)(f(x) - x^\top \nabla f(x)) - 2\omega f(x). \tag{35}$$

Since $f$ is convex, we have $f(x) - x^\top \nabla f(x) \leq 0$ for any $x \in \mathbb{R}^d$. Since $\omega = \gamma - 3\alpha \geq 0$, it follows that $\partial_t O + i[H, O] \leq 0$ and $f(x) \geq 0$ for all $x \in \mathbb{R}^d$, which implies that

$$\frac{\mathrm{d}}{\mathrm{d}t}\mathcal{E}(t) = \langle \partial_t \hat{O}(t) + i[\hat{H}(t), \hat{O}(t)] \rangle_t \leq 0.$$

$\square$

## C.3. Proof of Lemma 5

*Proof.* Similar to the proof of Lemma 2, we have

$$\frac{\mathrm{d}}{\mathrm{d}t}\mathcal{F}(t) = \langle \partial_t \hat{J}(t) + i[\hat{H}(t), \hat{J}(t)] \rangle_t. \tag{36}$$

Direct calculation shows that

$$\partial_t \hat{J}(t) + i[\hat{H}(t), \hat{J}(t)] = \mathcal{I}_1 + \mathcal{I}_2(t), \tag{37}$$

where

$$\mathcal{I}_1(t) = -2\omega x^\top \nabla f(x) + 2t(f(x) - x^\top \nabla f(x)) \leq 0, \tag{38}$$

$$
\begin{aligned}
\mathcal{I}_2(t) &= \beta t\left(\sum_{j=1}^{d}v_j^2\right) + \frac{\beta}{2}t^{5/2}\sum_{j,k=1}^{d}[v_j^2, \{A_k, x_k\}] \\
&= \beta t\left[\sum_{j=1}^{d}v_j^2 - 2\sum_{j,k=1}^{d}\left(\frac{\partial^2 f}{\partial x_j x_k}\right)x_k v_j\right] \\
&= \beta t\left(G(x) - x^\top \nabla G(x)\right) \leq 0.
\end{aligned}
\tag{39}
$$

The second equation uses commutation relation 5 in Lemma 3, and the last inequality is deduced from the convexity of $G(x)$. Combining (38) and (39), we prove the lemma. $\square$

## D. Technical details of complexity analysis

**Lemma 9.** *Assume that $f : \mathbb{R}^d \to \mathbb{R}$ is a $L$-Lipschitz function. For sufficiently smooth wave function $|\Phi\rangle$, we can prepare a quantum state $|\Psi\rangle$ such that*

$$\|\Psi - e^{-ihH_{k,2}}\Phi\| \leq \epsilon$$

*using*

$$\widetilde{\mathcal{O}}\left(\alpha dhL\right)$$

*queries to the first-order oracle $O_{\nabla f}$. Here, the $\widetilde{\mathcal{O}}(\cdot)$ notation suppressed poly-logarithmic terms in $1/\epsilon$.*

*Proof.* Recall that

$$H_{k,2} = \frac{\alpha}{2}\{-i\nabla, \nabla f\} = \frac{\alpha}{2}\sum_{j=1}^{d}\{p_j, v_j\}.$$

Note that this operator is independent of time and thus of $k$. To simulate the Hamiltonian $H_{k,2}$, we need to perform spatial discretization for the operators $p_j$ and $v_j$. The standard approach is to consider a $d$-dimensional regular mesh with $N$ grid points on each dimension, e.g., (An et al., 2022; Childs et al., 2022). The momentum operators can be implemented by applying Quantum Fourier Transform 2 times (with overall gate complexity $d \operatorname{poly} \log(N)$), as discussed in (Li et al., 2023). The discretized Hamiltonian operator takes the following form:

$$\widetilde{H}_{k,2} = \frac{\alpha}{2}\sum_{j=1}^{d}(\tilde{P}_j\tilde{V}_j + \tilde{V}_j\tilde{P}_j), \tag{40}$$

where $\|\tilde{P}_j\| \leq \mathcal{O}(N)$, and $\|\tilde{V}_j\| \leq L$, with $L$ the Lipschitz constant of $f$. By using $\mathcal{O}(1)$ queries to the first-order oracle $O_{\nabla f}$, we can implement a block-encoding of the matrix $\widetilde{H}_{k,2}$ with a normalization factor $a \leq \mathcal{O}(\alpha dNL)$, and an additional $\mathcal{O}(d)$ ancilla qubits Gilyén et al. (2019b, Lemma 29,30). With the block-encoded operator $H_{k,2}$, we can perform optimal Hamiltonian simulation by QSVT Gilyén et al. (2019b, Corollary 32). The total number of queries to the block-encoding is

$$\mathcal{O}\left(ah + \log(1/\epsilon)\right),$$

with an additional $\mathcal{O}(d(ah + \log(1/\epsilon)))$ elementary gates. Given that the input wave function $\Phi$ is sufficiently smooth, the discretization number $N$ can be chosen as $N = \operatorname{poly} \log(1/\epsilon)$ since the spatial discretization can be regarded as a pseudo-spectral method. It turns out that the overall query complexity of the Hamiltonian simulation reads $\widetilde{\mathcal{O}}(d\alpha hL)$, where the $\widetilde{\mathcal{O}}(\cdot)$ notation suppresses poly-logarithmic terms in $1/\epsilon$. $\square$

Now, we are ready to prove Theorem 6.

*Proof.* In Algorithm 1, each iteration requires the implementation of the quantum circuit

$$U_k = e^{-ihH_{k,1}}e^{-ihH_{k,2}}e^{-ihH_{k,3}}.$$

Note that $H_{k,1} = -\Delta/(2t_k^3)$ and the Laplacian operator $\Delta$ can be diagonalized by Fourier transform, so we can implement $e^{-ihH_{k,1}}$ using $\mathcal{O}(d\log^2(N))$ elementary gates. The Hamiltonian $H_{k,3}$ is a multiplicative operator with two commuting terms, i.e.,

$$e^{-ihH_{k,3}} = e^{-ih(\alpha^2+\beta)t_k^3\|\nabla f\|^2/2}e^{-ih(t_k^3+\gamma t^2)f}.$$

Since the functions $f$ and $\|\nabla f\|^2$ are multiplicative operators and reduce to diagonal matrices after spatial discretization. Therefore, the Hamiltonian $H_{k,3}$ is fast-forwardable and can be implemented using $\mathcal{O}(1)$ uses of the zeroth- and first-order oracle of $f$, respectively. Finally, by Lemma 9, the Hamiltonian $H_{k,2}$ can be simulated using $\widetilde{\mathcal{O}}(\alpha dhL)$ queries to the first-order oracle $O_{\nabla f}$. By iterating these steps for $K$ times, we can implement the quantum algorithm using $\mathcal{O}(K)$ queries to the zeroth-order oracle $O_f$ and $\widetilde{\mathcal{O}}(\alpha dhKL)$ queries to the first-order oracle $O_{\nabla f}$. $\square$

# E. Details of numerical experiments

## E.1. Numerical implementations of optimization algorithms

In the numerical experiments, we test four optimization algorithms: Stochastic Gradient Descent with momentum (SGDM), Nesterov's accelerated gradient descent (NAG), Quantum Hamiltonian Descent (QHD), and Gradient-based QHD. Our Python implementation of the numerical algorithms can be found in the supplementary materials.

**SGDM.** The iterative update rules for SGDM are as follows:

$$v_k = \eta_k v_{k-1} - (1 - \eta_k) s_k g_k,$$
$$x_k = x_k + v_k,$$

where $1 \leq k \leq K$ is the iteration number, $\eta_k$ is the momentum coefficient, $s_k$ is the step size, $g_k$ is an unbiased gradient estimator at $x_k$. We use

$$\eta_k = 0.5 + \frac{0.4k}{K}, \quad s_k = \frac{s_0}{k}.$$

with $s_0 = 0.01$, $v_0 = 0$, and a uniformly random initial guess $x_0$. The gradient estimator $g_k$ is obtained by adding a unit Gaussian random noise to the exact gradient $\nabla f(x_k)$.

**NAG.** The update rules of NAG are as follows:

$$x_k = y_{k-1} - s\nabla f(y_{k-1}),$$
$$y_k = x_k + \frac{k-1}{k+2}(x_k - x_{k-1}),$$

for $1 \leq k \leq K$. We choose $y_0 = 0$ and a uniformly random initial guess $x_0$. The step size is chosen as $s = 0.01$.

**QHD and gradient-based QHD.** Both QHD and gradient-based QHD are simulated following Algorithm 1. Note that QHD is a special case of gradient-based QHD with $\alpha = \beta = \gamma = 0$. The simulation is performed in a mesh grid with $N = 128$ grid points per dimension, with the momentum and kinetic operators implemented using FFT, as discussed in Appendix D. The step size varies with the test problems: We use $h = 0.01$ for the Styblinski-Tang and Michalewicz function, $h = 0.02$ for the Cube-Wave function, and $h = 0.005$ for the Rastrigin function.

## E.2. Non-convex test problems

The test problems used in this paper are defined as follows:

1. Styblinski-Tang function:
$$f(x, y) = 0.2 \times \left(x^4 - 16x^2 + 5x + y^4 - 16y + 5y\right),$$

   where we introduce a normalization factor of 0.2 for a better illustration. This function has a unique global minimizer at $(x^*, y^*) = (-2.9, -2.9)$, with the minimal function value $f(x^*, y^*) \approx -31.33$. The numerical algorithms are implemented over the square region $\{-5 \leq x, y \leq 5\}$.

2. Michalewicz function:
$$f(x, y) = -\sin(x)\sin(x^2/\pi)^{20} - \sin(y)\sin(2y^2/\pi)^{20}.$$

   This function has a unique global minimizer at $(x^*, y^*) = (2.2, 1.57)$, with the minimal function value $f(x^*, y^*) \approx -1.8$. The numerical algorithms are implemented over a square region $\{0 \leq x, y \leq \pi\}$.

3. Cube-Wave function:
$$f(x, y) = \cos(\pi x)^2 + 0.25x^4 + \cos(\pi y)^2 + 0.25y^4.$$

   This function has 4 global minima, namely, $(x^*, y^*) = (\pm 0.5, \pm 0.5)$. The minimal function value is $f(x^*) \approx 0.03$. The numerical algorithms are implemented over a square region $\{-2 \leq x, y \leq 2\}$.

4. Rastrigin function:
$$f(x, y) = x^2 - 10 \cos(2\pi x) + y^2 - 10 \cos(2\pi y) + 20.$$

This function has a unique global minimizer at $(x^*, y^*) = (0, 0)$, with the minimal function value $f(x^*, y^*) = 0$. The numerical algorithms are implemented over a square region $\{-3 \leq x, y \leq 3\}$.

