# OpenReview forum: "Quantum Optimization via Gradient-Based Hamiltonian Descent"
_ICML.cc/2025/Conference — ICML 2025 poster_

### Official Review · Reviewer_1Ka2 · 2025-03-12

**Overall Recommendation:** 4

**Summary:**

This paper proposed gradient-based Quantum Hamiltonian descent, which is motivated by insights from high-resolution differential equations and based on quantum Hamiltonian descent. They proved a faster convergence rate of gb-qhd under some reasonable assumptions and conduct numerical simulations that demonstrated the advantage of gb-qhd over original qhd.

-----
Update: I have read all the comments and rebuttals, and will take them into account for my evaluation of the paper.

**Claims And Evidence:**

Yes, the theoretical claims in the submission are supported by formal proofs, and performance claims are supported by numerical simulations.

**Essential References Not Discussed:**

There are not missing references.

**Experimental Designs Or Analyses:**

Yes, I have checked the validity of the numerical simulation results. They have provided details about experiments demonstrating gd-QHD's ability, which is a standard and fair comparison.

**Methods And Evaluation Criteria:**

Yes. For the theoretical part, the authors proved the convergence rate of their algorithms, which makes sense.
For the simulation part, the authors judged the performance by the function value with respect to iteration rounds, which also makes sense.

**Other Comments Or Suggestions:**

No more comments. The writing is very good.

**Other Strengths And Weaknesses:**

Strengths:
- Novel improvements for QHD in both theoretical and practical perspectives. They proposed a novel method called gd-QHD, and proved its convergence with better simulation results.
- The paper is well written in general, with tables and figures clearly demonstrating their results.

Weakness:
- A potential weakness is that the paper lacks experimental results on real devices. Besides numerical simulations, QHD provides further experiments on D-Waves and compared their experiment results with more classical algorithms, in many setups. Compared with that, this submission may be a little insufficient in this aspect. But I think the numerical simulations results in this paper is already convincing.

**Questions For Authors:**

- It seems that the authors have used a different notation from the original QHD paper, and this makes the comparison about the convergence rate (in the theoretical part) not so direct. Could the authors explain more about the convergence rates about both methods and make a comparison?

**Relation To Broader Scientific Literature:**

QHD opens a new quantum algorithm paradigm for designing quantum algorithms for optimization from a. physical viewpoint, and this work goes further on this direction. This could be beneficial for us to understand the relationship between quantum dynamics and optimization.

**Theoretical Claims:**

I do not check the proofs for the theoretical claims in detail, but I went through their proof ideas. They adopted a similar method (Lyapunov function) as the in the proof of QHD convergence. Therefore, I thought the claims are likely to be correct.

---

> ### Author Rebuttal · Authors · 2025-03-31
>
> We sincerely thank Reviewer 1Ka2 for their detailed comments and insightful suggestions. In particular, we appreciated the Reviewer's observation that our work could be beneficial to "understand the relationship between quantum dynamics and optimization".
>
> We address each of the Reviewer's questions as follows:
>
> 1. **Lack of experimental results on real devices**:
> We thank the reviewer for checking the feasibility of gradient-based QHD on real quantum devices such as D-Wave. Unlike vanilla QHD, implementing gradient-based QHD using analog simulators (e.g., D-Wave’s quantum computer) requires an explicit hardware encoding of the Hamiltonian $H_{k,2}\propto$ {$\nabla$, $\nabla f$}. This can be done efficiently for quadratic functions (not necessarily convex). For more sophisticated problems, e.g., higher-order polynomials, the encoding of $H_{k,2}$ must be evaluated on a case-by-case basis but remains feasible. We will include a brief discussion on the feasibility of an analog implementation of gradient-based QHD in the camera-ready version if this paper is accepted.
>
> 2. **Comparison with the original QHD paper**:
> The convergence rate of the original QHD is formulated in a more general form (Theorem 1 on page 21, [Leng et al., 2023](https://arxiv.org/abs/2303.01471)).
> $$\mathbb{E}[f(X_t)] - f(x^*) \le O(e^{-\beta_t}),$$
> where the time-dependent functions in QHD, i.e., $\alpha_t$, $\beta_t$, and $\gamma_t$, must satisfy the *ideal scaling condition*: $\dot{\beta}_t \le e^{\alpha_t}$, $\dot{\gamma}_t = e^{\alpha_t}$.
> Note that our choice of $\alpha, \beta,\gamma$ in this submission is irrelevant to the time-dependent functions in the original QHD paper. When we set $\alpha=\beta=\gamma=0$, our gradient-based QHD reduces to the vanilla QHD with $\alpha_t = -\log(t)$ and $\beta_t = \gamma_t = 2\log(t)$. In this case, they exhibit the same convergence rate $O(t^{-2})$.
> We will add this discussion to the camera-ready version if this paper is accepted.
>
> We sincerely appreciate the Reviewer's thoughtful feedback and constructive suggestions. Given our clarifications and the additional insights provided, **we hope the Reviewer might reconsider their evaluation and, if appropriate, adjust the score accordingly.**

---

> > ### Comment · Reviewer_1Ka2 · 2025-04-07
> >
> > Thank you for your detailed comments! I have another question after reading this rebuttal and the discussions from other reviewers. In Sections 6.2 and 6.3, you choose the special parameter $\beta = 0$, which reduces gradient-based QHD to a simpler form. It would be very helpful if you could clarify the following points:
> >
> > - From my understanding, setting $\beta = 0$ does not discard gradient information, as the gradients are already encoded into the $A_j$’s. Is that correct?
> > - I am a little confused by your second point. Does this mean that the parameters $\alpha$, $\beta$, and $\gamma$ in your submission are unrelated to the parameters $\alpha_t$, $\beta_t$, and $\gamma_t$ in the original QHD paper? If so, as you stated, choosing $\alpha = \beta = \gamma = 0$ reduces the gradient-based QHD to the vanilla QHD for a particular parameter setting. It does not seem immediate to me that gradient-based QHD is *always* a generalization of the original QHD for different parameter choices. Is that true?
> >
> > I appreciate your clarification on these points. Thank you!
> >
> > -------
> > Update: Thank you for your prompt reply! All my questions have been addressed.

---

> > > ### Author Response · Authors · 2025-04-07
> > >
> > > We sincerely thank Reviewer 1Ka2 for the constructive feedback. Below, we address the additional questions related to the gradient encoding and parameter setting in gradient-based QHD.
> > >
> > > > From my understanding, setting $\beta=0$ does not discard gradient information, as the gradients are already encoded into the
> > > $A_j$’s. Is that correct?
> > >
> > > Yes, the gradient is also included in $A_j = t^{-3/2}p_j + \alpha t^{3/2}v_j$, where $v_j = \partial_j f$. The gradients are encoded in the full Hamiltonian of gradient-based QHD as long as $\alpha \neq 0$ or $\beta \neq 0$.
> > >
> > > > Does this mean that the parameters $\alpha$, $\beta$, and $\gamma$ in your submission are unrelated to the parameters $\alpha_t$, $\beta_t$, and $\gamma_t$ in the original QHD paper? If so, as you stated, choosing $\alpha=\beta=\gamma=0$ reduces the gradient-based QHD to the vanilla QHD for a particular parameter setting. It does not seem immediate to me that gradient-based QHD is always a generalization of the original QHD for different parameter choices. Is that true?
> > >
> > > We confirm that the parameters $\alpha$, $\beta$, and $\gamma$ in this submission are different from the time-dependent functions $\alpha_t$, $\beta_t$, and $\gamma_t$ in the original QHD paper. In particular, the original QHD paper considers the Hamiltonian:
> > > $$H_1 = e^{\alpha_t - \gamma_t}(p^2/2) + e^{\alpha_t+\beta_t+\gamma_t}f(x).$$
> > > In this submission, we define the gradient-based QHD described by
> > > $$H_2 = (t^{-3/2}p + \alpha t^{3/2}v)^2 + \beta t^3 |\nabla f|^2/2 + (t^3 + \gamma t^2) f.$$
> > > Comparing these two Hamiltonians, it is clear that gradient-based QHD reduces to the original QHD if $\alpha = \beta = \gamma = 0$ and we choose $\alpha_t = -\log(t)$ and $\beta_t = \gamma_t = 2\log(t)$.
> > >
> > > In the current form, gradient-based QHD is the generalization of QHD under the specific choices: $\alpha_t = -\log(t)$ and $\beta_t = \gamma_t = 2\log(t)$.
> > > This leads to a simplified QHD formulation $H = p^3/(2t^3) + t^3 f$ that avoids distracting the audience with excessive hyperparameters. However, the general QHD ($H_1$) can be similarly extended to incorporate the gradient information (as in $H_2$). We will address this point in the camera-ready version if this paper is accepted.

---

### Official Review · Reviewer_F84Z · 2025-03-13

**Overall Recommendation:** 2

**Summary:**

This work proposes a gradient-based quantum hamiltonian descent (QHD), which generalizes the previously proposed based on function values. Theoretical and simulation results are also provided.

## After rebuttal

The authors clarified most of my concerns during the rebuttal. Hence, I increased my score. However, I still believe the clarity of the paper should be further improved. If the paper gets accepted as a final decision, I hope the authors can incorporate many aspects of the discussion in to the final version.

**Claims And Evidence:**

There are some claims that don’t' seem to align. For instance, (10) is only obtainable by assuming (9), in which case the RHS of (10) decays to 0. The next sentence reads: "Therefore, we expect the Hamiltonian dynamics … similar to that of high-resolution ODE." Then, in the next sentence, $\alpha, \beta, \gamma$ are NOT chosen according to (9). Please refer to the below sections for more details.

**Essential References Not Discussed:**

This work is in a fairly niche area, and I do not believe there is essential references to be discussed, other than the original QHD framework that the authors extensively refer to.

**Experimental Designs Or Analyses:**

This work requires simulating quantum dynamics with time dependent hamiltonian, which itself is not trivial. Moreover, in numerical experiments, none of the "theoretically motivated" choices are made, including the step size, the choice of $\alpha, \beta, \gamma$. I do not think the experimental results support the claim sufficiently.

**Methods And Evaluation Criteria:**

Not necessarily. For instance, in Section 6.1, the success probability is defined and is mentioned that the suboptimality measure ($\delta$) is set to 1. However, in Figure (3), all methods seem to start from (at k=0) initial suboptimality less than 1. I'm not sure how to interpret the plot.

**Other Comments Or Suggestions:**

Given that this is a machine learning conference, it would be nice to at least introduce some technical terms like "fast forwardable."

**Other Strengths And Weaknesses:**

- Classical optimization algorithms are quite sensitive to the step size. Executing all methods with the same step size of 0.2 does not tell much in terms of optimization, which is the aim of this work.
- It is hard to extract the main message of the paper. Theories appeal to high-resolution ODE, but then a lot of relaxations are made such that the theory the main text appeals to simply does not hold any more (e.g., $\beta=0$). Empirical evaluations are also not done rigorously (e.g., using the same step size of $0.2$ for all cases). I do not think, in the current form, this paper asserts something scientifically concrete to the readers.

**Questions For Authors:**

- What do you mean by NAG reduces oscillations in the optimization trajectory?  NAG is known to be more sensitive to noise, and actually oscillates more than gradient descent.
- What do you mean by "damped heavy ball motion"? (line 49)
- What is "iteration" for QHD in Fig 2? What/ is "success probability" of SGDM or NAG in Fig 2?
- Why do you start with (5)? Does (5) recover the Hamiltonian for original QHD with $\alpha = \beta = \gamma =0$? Or does this claim only hold for $\hat{H}(t)$?
- Line 300: gradient-based QHD Hamiltonian can be decomposed to three terms, where $H_{k,2}$ and $H_{k,3}$ vanishes with $\alpha = \beta = \gamma  = 0$. So just simulating $H_{k,1}$ will recover QHD? Then why simulate the other two terms? Wouldn't that necessarily incur higher qubit/gate complexity?
- Moreover, in Sec 5.2, it's mentioned that $H_{k,1}$ is fast forwardable. So vanilla QHD is fast forwardable?
- In Section 6.1, the success probability is defined and is mentioned that the suboptimality measure (\delta) is set to 1. However, in Figure (3), all methods seem to start from (at k=0) initial suboptimality less than 1. What is going on?
- Why is $\beta=0$ in Sec 6.2? Based on (12), that removes the gradient component, which seems to directly contradict what the paper asserts.

**Relation To Broader Scientific Literature:**

Given my above points, I am not sure what is the key contribution of this paper to the broader scientific literature.

**Theoretical Claims:**

While theoretical results are presented, they do not seem to align with the algorithm. For global convergence, for instance, $\beta=0$ in which case the gradient component of the Hamiltonian disappears. Theoretically, what is the benefit of gradient-based QHD over vanilla QHD?

---

> ### Author Rebuttal · Authors · 2025-03-31
>
> We sincerely thank Reviewer F84Z for their detailed comments and insightful suggestions.
>
> First, we would like to clarify the primary contribution of this submission, as it appears to have been misinterpreted by Reviewer F84Z.
>
> The primary objective of this work is to propose a *novel* quantum Hamiltonian-based algorithm (gradient-based QHD) for continuous optimization. While our Hamiltonian is inspired by the Bregman Lagrangian and the high-resolution ODE framework, it is neither equivalent to them nor a direct extension. Instead, it possesses unique structures and properties. In this work, we study the dynamical and algorithmic properties of gradient-based QHD independently using new mathematical tools. This distinction was made clear in the original submission:
> > In this work, we do not limit our choice of the parameters $\alpha$, $\beta$, and $\gamma$ ... explore a larger family of dynamical systems for continuous optimization problems.
>
> **We now address the Reviewer's major concerns:** (minor points ignored due to page limit)
>
> > There are some claims that don’t seem to align ... are NOT chosen according to (9).
>
> The Hamiltonian dynamics in (10) serve to formally illustrate the connection between gradient-based QHD and high-resolution ODEs. Gradient-based QHD does not reduce to high-resolution ODEs, and its convergence properties have been established independently (Theorems 1 & 4). **We believe our claims are self-consistent and supported by both the theoretical and numerical evidence presented in the original submission.**
>
> > In Section 6.1, the success probability is defined ... I'm not sure how to interpret the plot.
>
> Figure 3(a) depicts the expectation value of the sub-optimality gap (caption: "function value"). It is important to note that the average sub-optimality gap is not equivalent to the success probability measure defined in Section 6.1: even if the initial average sub-optimality is below 1, this does not imply that all solutions achieve an optimality gap lower than 1.
> Additionally, we have identified a typo in the caption of Figure 3(b): "Success probability" should be corrected to "Gradient norm." We sincerely apologize for any confusion and will ensure that this typo is corrected in the camera-ready version if the paper is accepted.
>
> > ... the gradient component of the Hamiltonian disappears ... what is the benefit of gradient-based QHD over vanilla QHD?
>
> We thank the Reviewer for the question regarding the benefits of gradient-based QHD over vanilla QHD. By definition (Eqs. (12)–(13)), the gradient appears in both $\frac{1}{2}\sum^d_{j=1}A_j$ and $\beta t^3\|\nabla f\|^2$. Therefore, even if we set $\beta = 0$, gradient information remains present in $A_j$ as long as $\alpha \neq 0$. Theoretically, the inclusion of the gradient in the Hamiltonian results in a larger spectral gap compared to vanilla QHD. A sufficiently large spectral gap is crucial for the success of Hamiltonian-based optimization algorithms, as is well understood in the context of adiabatic algorithms and vanilla QHD.
>
> > ... in numerical experiments, none of the "theoretically motivated" choices are made, including the step size, ...
>
> As long as the parameters $\alpha, \beta, \gamma$ satisfy the conditions in Theorem 1, our result holds independent of specific step-size choices. Our choice of the parameters in the numerical experiment also aligns with the conditions discussed in Theorem 1.
>
> > ...proof of Theorem 6 ... asserts $H_{k,3}$ can be implemented in constant time. What is the reasoning?
>
> The Hamiltonian $H_{k,3}$ represents a point-wise multiplication of a function to the wave function, and its spatial discretization directly leads to a diagonal operator acting on the (discretized) wave function. Since all the diagonal elements of the discretized $H_{k,3}$ are efficiently computable via the query access to $f$ and $\nabla f$, we can simulate $e^{itH_{k,3}}$ using $O(\log(t))$ gates.
>
> > Executing all methods with the same step size of 0.2 does not tell much in terms of optimization, which is the aim of this work.
>
> In our preliminary experiments, we implemented the test in Section 6.2 with a range of step sizes ($h \in [0.05, 0.5]$), and we always observed similar convergence behavior. Therefore, to maintain consistency, we fix $h = 0.2$ in the submission. We will add all these results in the camera-ready version if this paper is accepted.
>
> > Line 300: ... simulating $H_{k,1}$ will recover QHD? ... $H_{k,1}$ is fast forwardable. So vanilla QHD is fast forwardable?
>
> When setting $\alpha=\beta=\gamma=0$, we have
> $$H_{k,1} = -\Delta/(2t^3), \quad H_{k,3} = t^3 f,$$
> and $H_{k,1}+H_{k,3}$ recovers QHD.
> Clearly, $H_{k,3}$ does not vanish, and just simulating $H_{k,1}$ will not recover QHD.
> Moreover, since $H_{k,1}$ and $H_{k,3}$ can not be simultaneously diagonalized, QHD is not fast-forwardable.
>
> **We appreciate the Reviewer's time and consideration and hope this clarification helps in reassessing our submission.**

---

> > ### Comment · Reviewer_F84Z · 2025-04-08
> >
> > Thank you for your detailed reply. I mainly disagree with the following statement:
> >
> > > Gradient-based QHD does not reduce to high-resolution ODEs, and its convergence properties have been established independently (Theorems 1 & 4). We believe our claims are self-consistent and supported by both the theoretical and numerical evidence presented in the original submission.
> >
> > I believe the fundamental flaw is that it is unclear what algorithm Theorems 1 and 4 analyze. Is it Algorithm 1? But isn't Algorithm 1, which to be precise is the proposed method, a time-discretized version of Eq (14)? QHD fundamentally is a quantum-dynamics simulation-based algorithm. Theorems 1 and 4 assume (14) can be simulated exactly. Is that possible?
> >
> > To be more specific, how can (17) be implemented exactly? How should I interpret the $\approx$ sign? It will necessarily incur some error; please correct me if I misunderstand anything. Due to this reason, I find Theorem 6 to be more aligned with the theoretical analysis of the performance of the proposed method.
> >
> > Further, I'm a bit confused by the decomposition of $\hat{H}(t_k)$ above Eq. (17). There, it seems that the gradient norm term from $H_{k, 3}$ persists even with $\beta=0$ (the case Theorems 1 and 4 analyze). This does not seem to align with Eq. (14), which, to my understanding, Algorithm 1 is implementing (in the time-discretized version).
> >
> > In Theorem 6, the query complexity depends linearly on $\alpha$. How do I interpret this result if $\alpha=0$? I don't see any problem with setting $\alpha=0$ in Theorem 1, as it is written; but clearly I cannot plug in $\alpha=0$ in Theorem 6. Further, the query complexity in Theorem 6 also depends linearly on the step size $h$, which to me reads that I should use the smallest possible step size to minimize the query complexity, which sounds counter-intutivie. So, in general, I'm confused about what algorithm Theorems 1 and 4 analyze, and I see many disconnections between them and Theorem 6, which, to my understanding, analyzes Algorithm 1, which is the proposed algorithm, not Eq (14).
> >
> > Lastly, in the proof of Theorem 6, many steps seem handwavy. To be specific, what do you mean by $\Phi$ being "sufficiently" smooth? How smooth should it be to have $N = \text{poly} \log (1/\epsilon)?$ How does the spatial discretization being regarded as a pseudo-spectral method "turns out" that the overall query complexity simply reads $\tilde{O} (d \alpha h L)$?
> >
> > To sum up, I find the submitted manuscript requires significant clarification, and I find the theoretical contribution to be not rigorous and the empirical evaluation not extensive. Therefore, I keep my original score.

---

> > > ### Author Response · Authors · 2025-04-08
> > >
> > > We sincerely thank Reviewer F84Z for the informative comment and feedback. Below, we address the additional questions related to the implementation and interpretation of gradient-based QHD.
> > >
> > > ---
> > >
> > > **Comment 1: discrepancy between Theorems 1 & 4 and Algorithm 1.**
> > >
> > > Theorems 1 and 4 analyze the convergence rate of the continuous-time gradient-based QHD dynamics (generated by the Hamiltonian as in (12)). Theorem 6 gives a rigorous complexity analysis of Algorithm 1 (i.e., time discretization of gradient-based QHD). It is worth noting that:
> > > 1. The continuous-time dynamics itself is **not a quantum algorithm**, but a mathematical model of the proposed quantum algorithm (Algorithm 1); and
> > > 2. We **never claimed that Algorithm 1 is a perfect simulation of the continuous-time dynamics**; instead, the interesting message from our numerical results (Section 6) is that Algorithm 1 converges with "relatively large" step sizes (e.g., $h = 0.2$). This strongly suggests that **the convergence of Algorithm 1 still happens without perfectly simulating the continuous-time dynamics**.
> > >
> > > At first glance, this may seem miraculous—or even counterintuitive. However, the observation is entirely consistent with our experience from classical gradient descent (GD). The continuous-time limit of GD (i.e., $x_k = x_{k-1} - h \nabla f(x_{k_1})$) corresponds to the gradient flow: $\dot{X}_t = - \nabla f(X_t)$. For convex $f$, gradient flow achieves a convergence rate of $f(X_t) - f(x^*) \le O(1/t)$ for convex $f$. Meanwhile, gradient descent converges whenever the step size satisfies $h \le 1/L$, where $L$ is the Lipschitz constant of $\nabla f$. This indicates that GD, as a discrete algorithm, converges without perfectly simulating the gradient flow. In our paper, we (numerically) confirm that a similar phenomenon holds in the quantum setting.
> > >
> > > Ideally, we would like to establish the convergence of Algorithm 1 independently of the continuous-time dynamics—just as the convergence of GD can be proven without relying on the analysis of gradient flow. However, this remains an open problem, as we have not yet identified suitable analytical tools. Since the current manuscript already includes a continuous-time convergence analysis and strong numerical evidence demonstrating the effectiveness of the discrete-time algorithm, we believe our results are on par with the standards of the machine learning community. We leave the technical analysis (i.e., a rigorous proof of the convergence of Algorithm 1 for non-zero $h$) for future work.
> > >
> > > ---
> > >
> > > **Comment 2: decomposition of $\hat{H}(t_k)$**
> > >
> > > The gradient norm term in $H_3$ comes from the expansion of the first term in the Hamiltonian: $A^2_j = (t^{-3/2}p + \alpha t^{3/2}v_j)^2$ in (12). Direct calculation shows that this will add a $\alpha^2 t^3 v^2_j$ term to the "diagonal Hamiltonian" (i.e., $H_3$).
> > >
> > > ---
> > >
> > > **Comment 3: query complexity depends linearly on $\alpha$**
> > >
> > > In the extreme case where we set $\alpha = 0$, no gradient appears in the Hamiltonian. This means that the Hamiltonian can be simulated without querying $\nabla f$ when $\alpha = 0$, therefore, the query complexity (to $\nabla f$) becomes 0. **This is very natural and intuitive, as reflected by our Theorem 6.** Note that the query complexity to $f$ is unchanged no matter what values are chosen for $\alpha$ and $\beta$; therefore, there is no free lunch in Algorithm 1 even if we eliminate the gradient component (and the dynamics essentially reduce to a close variant of the original QHD).
> > >
> > > ---
> > > **Comment 4: the proof of Theorem 6**
> > >
> > > We apologize for omitting some details in the proof of Theorem 6, but we do not think it affects the correctness of our proof. The pseudo-spectral method (i.e., DFT for Laplacian, regular quadrature for potential) is the go-to real-space simulation algorithm for Schrodinger equations. When the wave function has Fourier coefficients that decay super-polynomially (indicating the wave function is "smooth" or $C^\infty$), it is sufficient to use a truncation number $N = \mathrm{poly}\log(1/\epsilon)$. Based on this spatial discretization, we derived the $\tilde{O}(d\alpha h L)$ query complexity. We will add a detailed discussion on the spatial discretization of the Hamiltonian in the camera-ready version if this paper is accepted.
> > >
> > > ---
> > >
> > > **We appreciate the Reviewer's time and consideration and hope this clarification helps in reassessing our submission.**

---

### Official Review · Reviewer_NP8c · 2025-03-13

**Overall Recommendation:** 3

**Summary:**

In this submission, the authors presented a variant of the prominent quantum Hamiltonian descent (QHD) algorithm by adding the help of the gradient information. More specifically, the authors proposed a new time-dependent Hamiltonian as in Eq (4) which, unlike the original QHD, contains the gradient information. The authors proved the convergence of the new method. More surprisingly, the authors presented numerical results showing the gradient-based QHD outperforms the original QHD in many different settings.

**Claims And Evidence:**

Theoretical proofs and numerical evidence are provided for the claims.

**Essential References Not Discussed:**

N/A

**Experimental Designs Or Analyses:**

The numerical experiments are sound and valid.

**Methods And Evaluation Criteria:**

The results are backed by numerical evaluation.

**Other Comments Or Suggestions:**

1. Page 4: classical QHD -> original QHD. The term "classical QHD" might be misleading since readers might think it is referring to a classical algorithm.

2. The paper arXiv:2410.14243 might prove a better algorithm for simulating time-dependence Hamiltonians.

**Other Strengths And Weaknesses:**

I think this submission has made a solid contribution to quantum machine learning and optimization. It is a nice extension of the original QHD. The only weakness I can see is the lack of the convergence rate comparison with the original QHD.

**Questions For Authors:**

What is the intuition of the Lagrangain function in Eq (5)?

**Relation To Broader Scientific Literature:**

The problems this submission studies may find applications in quantum machine learning.

**Theoretical Claims:**

I checked the proofs and they appear to be correct.

---

> ### Author Rebuttal · Authors · 2025-03-31
>
> We sincerely thank Reviewer NP8c for their detailed comments and insightful suggestions. In particular, we appreciate the Reviewer's observation that our submission "makes a solid contribution to quantum machine learning and optimization."
>
> Below, we address each of the Reviewer's questions individually.
>
> 1. **Lack of the convergence rate comparison with the original QHD**:
> The convergence rate of the original QHD is formulated in a more general form (Theorem 1 on page 21, [Leng et al., 2023](https://arxiv.org/abs/2303.01471)).
> $$\mathbb{E}[f(X_t)] - f(x^*) \le O(e^{-\beta_t}),$$
> where the time-dependent functions in QHD, i.e., $\alpha_t$, $\beta_t$, and $\gamma_t$, must satisfy the *ideal scaling condition*: $\dot{\beta}_t \le e^{\alpha_t}$, $\dot{\gamma}_t = e^{\alpha_t}$.
> Note that our choice of $\alpha, \beta,\gamma$ in this submission is irrelevant to the time-dependent functions in the original QHD paper. When we set $\alpha=\beta=\gamma=0$, our gradient-based QHD reduces to the vanilla QHD with $\alpha_t = -\log(t)$ and $\beta_t = \gamma_t = 2\log(t)$. In this case, they exhibit the same convergence rate $O(t^{-2})$. We will add this discussion to the camera-ready version if this paper is accepted.
>
> 2. **classical QHD -> original QHD**: We thank the Reviewer for this thoughtful suggestion and would be happy to incorporate this change in the camera-ready version if the submission is accepted.
>
> 3. **Better quantum algorithms for time-dependent Hamiltonian simulation**: We will include the reference arXiv:2410.14243, along with several other results on commutator scaling, in the camera-ready version.
>
> 4. **The intuition of the Lagrangain function in Eq (5)**:
> Eq. (5) is our Lagrangian design that is inspired by both the Bregman Lagrangian and the high-resolution ODE.
> Specifically, the convergence analysis of the high-resolution ODE (Shi et. al.) leverages a Lyapunov function
> $$\frac{d \mathcal{E}(t)}{d t} \le - \left[\sqrt{s}t^2 + \left(\frac{1}{L} + \frac{s}{2}\right)t + \frac{\sqrt{s}}{2L}\right]\|\nabla f(X)\|^2 < 0.$$
> The Lyapunov function can be interpreted as a form of system energy that includes $\nabla f$, which motivates our design of (5).
>
> We sincerely appreciate the Reviewer's thoughtful feedback and constructive suggestions. Given our clarifications and the additional insights provided, **we hope the Reviewer might reconsider their evaluation and, if appropriate, adjust the score accordingly.**

---

### Official Review · Reviewer_Wa1N · 2025-03-17

**Overall Recommendation:** 2

**Summary:**

This paper explores quantum algorithms for solving unconstrained optimization problems. Given that Nesterov's accelerated gradient descent admits a classical Hamiltonian dynamics interpretation, it is natural to consider leveraging quantum Hamiltonian dynamics for algorithm design. In particular, Leng et al. proposed the Quantum Hamiltonian Descent (QHD) algorithm, which defines a quantum evolution via a time-dependent Schrödinger equation. In QHD, the potential term is proportional to the objective function $f$ and increases with time $t$, while the kinetic energy term decreases with $t$.

Building on the intuition from the high-resolution ODE framework by Shi et al., this work extends QHD by incorporating a gradient term of the objective function $f$ into the potential. The resulting algorithm is called gradient-based QHD. The authors then proved a convergence guarantee of gradient-based QHD, developed a quantum algorithm that simulates discrete-time gradient-based QHD, and conducted numerical experiments testing the performance of gradient-based QHD.

**Claims And Evidence:**

1. Gradient-based QHD converges to a global minima of the objective function $f$ with an inverse quadratic convergence rate, and converges to a stationary point with the same convergence rate.
2. In some cases, gradient-based QHD yields solution that are an order of magnitude better than those obtained by other methods.

Both claims appear to be correct; however, I have some concerns about their implications. Please find them below.

**Essential References Not Discussed:**

Not anything in particular I can think of

**Experimental Designs Or Analyses:**

The experimental setups and results are generally well-presented. However, the scale of these examples, particularly the dimension of the objective function, appears relatively small compared to the instances studied in the QHD paper by Leng et al. Additionally, Leng et al. demonstrated an analog implementation of QHD on the D-Wave system, which can be more efficiently executed on near-term quantum devices than digital implementations. In contrast, this paper does not provide an analogous implementation for gradient-based QHD.

Moreover, I have the following questions regarding the setups:
1. What is the advantage of discretizing gradient-based QHD rather than approximating its continuous dynamics using Hamiltonian simulation algorithms, as done in the QHD paper?
2. The computation in each iteration of gradient-based QHD appears significantly more complicated than in NAG, particularly due to the presence of the Laplacian operator. Is it fair to compare their performance based on the same number of iterations?

**Methods And Evaluation Criteria:**

Please refer to the following two parts.

**Other Comments Or Suggestions:**

Typo in line 096: quantum Hamiton descent -> quantum Hamiltonian descent

**Other Strengths And Weaknesses:**

I don't have further comments

**Questions For Authors:**

I don't have further questions other than the existing ones above.

**Relation To Broader Scientific Literature:**

This paper proposes a novel idea and enriches the literature on solving unbounded continuous optimization problems using quantum Hamiltonian dynamics.

**Theoretical Claims:**

The proofs for the convergence rates appear correct to me, and the techniques are quite elegant. However, I am concerned about potential overhead hidden in the big-O notation—possibly involving factors related to the dimension of the objective function or the mass of the initial wave function in regions where the function value is small. Given that, in the worst case, finding the global optimum is NP-hard, assuming P$ \neq$NP, such overhead could be superpolynomially large, making an inverse-quadratic convergence rate insufficient. While a similar issue exists in the convergence analysis of QHD by Leng et al., their convergence rate for convex functions is inverse-exponential, which strongly suggests that the required simulation time remains polynomially bounded in the worst case. Therefore, I believe the authors need to provide further justification for why an inverse-quadratic convergence rate is meaningful.

Minor question: In Theorem 6, why is a separate quantum first-order oracle necessary, given that Jordan's gradient estimation algorithm allows us to compute the gradient using a constant number of queries to a quantum function value oracle?

---

> ### Author Rebuttal · Authors · 2025-03-31
>
> We sincerely appreciate Reviewer Wa1N's thorough feedback and valuable insights. In particular, we thank the Reviewer for recognizing our techniques as "elegant" and acknowledging that this work "proposes a novel idea and enriches the literature on solving unbounded continuous optimization problems." We address each of the Reviewer's questions below:
>
> 1. **Potential overhead hidden in the big-O notation**:
> According to our proof of Theorem 1, the detailed convergence rate is (for some $0 < T_0 \le 1/\alpha$):
> $$\mathbb{E}[f(X_t)] \le \frac{K_0 + D_0}{t^2 + (\gamma - 3\alpha) t},\quad 0 < T_0 \le t,$$
>   - $K_0 = \langle \Psi(T_0)|(-\Delta)|\Psi(T_0)\rangle / T^4_0$: the initial kinetic energy. Independent of $f$ and typically scales as $O(d)$, e.g., for a standard Gaussian $\Psi_{0}$.
>   - $D_0 = \mathbb{E}\left[\|\nabla f(X_{T_0})\|^2+4\|X_{T_0}\|^2+(T^2_0+\omega T_0)f(X_{T_0})\right]$: generally scales as $O(d)$ due to $\|\nabla f\|^2$.
> Therefore, there might be an additional $O(d)$ overhead in our result. We will add this discussion to the camera-ready version if this paper is accepted.
>
> 2. **NP-hardness of global minimization & justification for inverse-quadratic convergence**:
>   - The inverse-quadratic convergence rate (Theorem 1) is established for general convex $f$. While finding the global optimum of a non-convex objective function is in general NP-hard, for the problem class of interest (i.e., general convex optimization), there exist polynomial-time classical algorithms with query complexity $O(n^2$) [Lee et al., COLT '18]( https://proceedings.mlr.press/v75/lee18a/lee18a.pdf). There is no ``super-polynomial overhead'' for the problem class we discussed in Theorem 1.
>   - The $O(t^{-2}$) rate is known to be optimal for classical first-order methods. Although a direct quantum counterpart has not yet been established, strong evidence suggests that there is no quantum speedup for generic convex optimization (e.g., [Garg et al., 20](https://arxiv.org/abs/2010.01801)). Our convergence rate may be already near optimal.
>   - Additionally, we emphasize that our numerical results demonstrate a significant advantage of gradient-based QHD for non-convex optimization. This observed performance extends beyond the scope of Theorem 1. To further clarify this distinction, we will explicitly highlight the convexity assumption in Theorems 1 & 4 in the camera-ready version.
>
> 3. **Necessity of quantum first-order oracle in Theorem 6**:
> We agree with the Reviewer that the requirement for a quantum first-order oracle $O_{\nabla f}$ can potentially be eliminated by Jordan's algorithm. However, the query complexity for obtaining an $\epsilon$-approximate gradient scales as $\mathcal{O}(\sqrt{d}/\epsilon)$ without a strong smoothness characterization of $f$ ([Gilyen et al., SODA'19](https://epubs.siam.org/doi/abs/10.1137/1.9781611975482.87)). In this work, we focus on the general convergence properties and leave the integration of quantum gradient estimation for future study.
>
> 4. **Numerical experiments are low-dimensional & analog implementation of QHD**:
> We thank the reviewer for highlighting the feasibility of gradient-based QHD on near-term quantum devices. Unlike vanilla QHD, implementing gradient-based QHD using analog simulators requires an explicit hardware encoding of the Hamiltonian $H_{k,2}\propto$ {$\nabla$, $\nabla f$}. This can be done efficiently for quadratic functions (not necessarily convex). For more sophisticated problems, e.g., higher-order polynomials, the encoding of $H_{k,2}$ must be evaluated on a case-by-case basis but remains feasible. We will include a brief discussion on this point in the camera-ready version if this paper is accepted.
>
> 5. **Advantage of discretizing gradient-based QHD**: The discretization method proposed in this submission utilizes the Trotter product formula, which can be viewed as a quantum simulation algorithm. Our approach, based on the product formula, has a simple structure and is potentially more straightforward to implement.
>
> 6. **Fairness in comparing gradient-based QHD and NAG based on the same number of iterations**: We agree with the Reviewer that the iteration steps in gradient-based QHD are more complicated.
> According to the proof of Lemma 9 and Theorem 6, the query and gate complexity of each iteration in gradient-based QHD scale as $\tilde{O}(d)$. In contrast, while each iteration of NAG requires only a single query to $\nabla f$, its time complexity remains $O(d)$.
> Therefore, in terms of actual runtime, gradient-based QHD is asymptotically comparable to NAG, making our comparison based on the same iteration count fair.
>
> 7. **Typo in line 096**: we have corrected the typo.
>
> We sincerely appreciate the Reviewer's thoughtful feedback and constructive suggestions. Given our clarifications and the additional insights provided, **we hope the Reviewer might reconsider their evaluation and, if appropriate, adjust the score accordingly.**

---

### Decision · Program_Chairs · 2025-05-01

**Decision:**

Accept (poster)

**Comment:**

This paper proposes a quantum algorithm for minimizing differentiable functions. The algorithm is a direct time discretization of quantum Hamiltonian dynamics associated with a novel Hamiltonian. While the convergence of the continuous-time dynamics has been established, the proposed algorithm is a relatively straightforward discretization, and its step size selection rule and corresponding convergence have not been rigorously discussed. Experimental results demonstrate the superiority of the proposed algorithm.

Although there are several obvious gaps, the ideas and some of the results in this paper, such as the proposed Lagrangian and the condition in Theorem 4, appear interesting and worthy of discussion. Moreover, the remaining concerns of Reviewer F84Z, who gave the lowest score, pertain to the presentation, which can be easily addressed. Therefore, I recommend accepting this paper.

In the revision, please discuss the advantages over quantum Hamiltonian descent and provide an interpretation of the Lagrangian. Additionally, please make it explicit that Theorem 1 assumes convexity of the objective function; the current presentation is confusing. Please also address Reviewer Wa1N’s concerns, as done in the author rebuttal, in the revised version.